# Merizo: a rapid and accurate protein domain segmentation method using invariant point attention

Andy M. Lau[1], Shaun M. Kandathil [1] & David T. Jones [1] ✉

The AlphaFold Protein Structure Database, containing predictions for over 200 million proteins, has been met with enthusiasm over its potential in enriching structural biological research and beyond. Currently, access to the database is precluded by an urgent need for tools that allow the efficient traversal, discovery, and documentation of its contents. Identifying domain regions in the database is a non-trivial endeavour and doing so will aid our understanding of protein structure and function, while facilitating drug discovery and comparative genomics. Here, we describe a deep learning method for domain segmentation called Merizo, which learns to cluster residues into domains in a bottom-up manner. Merizo is trained on CATH domains and fine-tuned on AlphaFold2 models via self-distillation, enabling it to be applied to both experimental and AlphaFold2 models. As proof of concept, we apply Merizo to the human proteome, identifying 40,818 putative domains that can be matched to CATH representative domains.

Domains are locally compact regions within proteins that can fold independently of the rest of the protein and can sometimes support a biological function on their own. The fold of a domain is not unique to individual proteins but can be found and adopted by a variety of different sequences. Structural domains are well-annotated in databases such as CATH[1,2], ECOD[3], Pfam[4] and SCOP[5], which leverage sequence, structure, function and their evolutionary relationships to provide a comprehensive hierarchical classification of fold space, each with different levels of granularity.

A long-standing challenge in structural biology is the problem of domain segmentation, or more precisely, how to divide protein structures into their constituent domains. Wetlaufer envisioned splitting proteins into domains as early as 1973, but even the denominations of what constitutes a domain are contested by different classification databases[6]. The structure of protein kinase CK2 (PDB 3BQC [https://doi.org/10.2210/pdb3BQC/pdb] chain A) for example, is classified in CATH as a two-domain protein (superfamilies 3.30.200.20 and 1.10.510.10), but in ECOD as a single domain (ECOD 206.1.1.24)[7]. The difference in the assignment is due to ECOD preserving an active site formed between the N- and C-terminal lobes, while CATH bases its assignment on the internal structures of the two (sub)domains.

Early segmentation methods such as PUU[8], DOMAK[9] and DETECTIVE[10] published in the 1990s relied on the proposition that domains have a high intra-to-inter-domain contact ratio, and directly applied this principle to each protein structure to identify its domains. Newer methods, such as those used in automatic domain classification by CATH, ECOD and SCOP, instead capitalise on the extensive annotations already conducted and use existing classifications to seed and find similar domains in query structures based on various criteria[7]. CATH, for example, uses CATHEDRAL which clusters new structures to already assigned domains by detecting similarities between the secondary structure components in the protein core, using a graph theory-based algorithm[11].

Broadly, methods that identify domains can be divided into two groups based on how segmentation is conducted. PUU, DOMAK and DETECTIVE, as well as the more recent DeepDom[12], DistDom[13] and FUPred[14], all conduct segmentation in a top-down fashion, in which the most likely "cut points" along the protein sequence are determined and used to partition it into domains. A key disadvantage of this regime is that discontinuous domains - those that fold in 3D space via two or more disjoint stretches of residues are typically left over-segmented as separate domains. The dual of the task, and a more challenging one, is

[1]Department of Computer Science, University College London, London WC1E 6BT, UK. ✉e-mail: d.t.jones@ucl.ac.uk

to instead predict the domain membership of each residue individually. The second category of domain detection methods is therefore composed of bottom-up methods such as UniDoc[15], SWORD[16] and DomBPred[17], which decompose the input protein into fragments that are then progressively aggregated into domains. SWORD, in particular, proposes several alternative configurations as well as an optimal one, which can be reviewed by users to identify a suitable partitioning.

The AlphaFold Protein Structure Database (AFDB) contains predicted models for over 200 million protein sequences and constitutes a valuable expansion of protein space[18]. An obvious use case for domain segmentation is in the high-throughput identification of domains from the AFDB, facilitating their sorting into structure databases such as CATH. Compared to experimentallyderived PDB structures, AFDB models may exhibit less optimal packing or folding, particularly for rare folds with limited known sequence homology. Furthermore, unlike experimental structures, the generation of in silico models is not constrained by the same factors (such as crystallisation success, quality, etc.), enabling the modelling of the entire sequences, including previously difficult-to-resolve regions. As such, many AFDB models also feature long stretches of unstructured regions that may hinder the performance of methods that are not prepared to operate on such models. More recently, methods such as DPAM have been developed to specifically operate on AFDB models, leveraging a combination of inter-residue distances, structural similarity to ECOD domains and the predicted aligned error (PAE) map produced by AlphaFold2 to inform domain assignment[19].

Our approach to the domain segmentation problem, called Merizo, is based on a deep neural network which conducts bottom-up domain assignment by learning to directly cluster residues into domains, based on a combination of its sequence and structure. Notably, our method makes use of the Invariant Point Attention (IPA) module introduced in AlphaFold2[20], leveraging its ability to mix together sequence and coordinate information to directly encode a protein structure into a latent representation (Fig. 1). Residue embeddings are clustered together by using an affinity learning[21–23] approach whereby the ground-truth domain map is used directly as an objective, thereby allowing class index-invariant predictions. Residues that are part of the same domain are encouraged towards the same embedding, while those that are not are encouraged to have different embeddings. Merizo is trained on CATH domain annotations and finetuned on a subset of AFDB models using a self-knowledge distillation approach, allowing the network to be equally applied to experimental structures as well as those generated by AlphaFold2. Furthermore, we show how fast and accurate methods such as Merizo can be applied to the human genome, identifying 40,818 putative domains that can be matched to existing structures in CATH at various levels of similarity.

## Results

### Benchmark against existing state-of-the-art methods

Merizo was trained on 17,287 multi-domain proteins with annotations sourced from CATH 4.3, and another 663 chains were held out to be used as a testing set (referred to as CATH-663). Targets in CATH-663 do not share any domains from the same homologous superfamily as the training set in order to better gauge performance on folds that the network has not seen before. How well a predicted assignment agrees with the ground truth can be quantified via a number of different measures. Here, we score predictions based on (1) how well the residues in a predicted domain overlap with a true domain, measured via the intersect-over-union (IoU) between residues in the predicted and ground-truth domain, and (2) how precise the predicted domain boundaries are, when assessed using the Matthews Correlation Coefficient (MCC; Supplementary Methods). The MCC describes the correlation between the predicted and ground-truth boundary positions, and a boundary is deemed correct if it is predicted within $\pm m$ residues of a ground-truth domain boundary (where $m$ is evaluated at 20

residues). Both scores are calculated at the domain level, and we report the domain length-weighted average for each target.

Our benchmark compares the accuracy of domain assignments by Merizo against those produced by four recently published methods including DeepDom[12], a CNN-based method from Eguchi et al[24]. (referred to as Eguchi-CNN), SWORD[16] and UniDoc[15]. Both DeepDom and Eguchi-CNN are machine learning (ML)-based methods and operate on primary sequence and distance map inputs respectively. In contrast, SWORD and UniDoc are non-ML-based and conduct segmentation on coordinates in a bottom-up fashion by clustering low-level structural elements into domains, in a manner similar to Merizo. In addition to the four published methods, we include four baseline measures, including scoring ECOD assignments against CATH (where ECOD assignments are treated as a prediction result), and three random assignment methods prefixed with 'Random', where the domain count is estimated according to the Domain Guess by Size method[25]. Targets are then divided into either equally or unequally sized segments ('Random equal/unequal') or each residue is assigned into a domain at random ('Random assigned').

A summary of the benchmark is shown in Fig. 2. Overall, it can be seen that Merizo is the most performant method on the CATH-663 set when scoring by IoU, achieving a similar median IoU to the ECOD baseline. Merizo is followed closely by UniDoc which exhibits a similar median IoU, albeit with a wider distribution. As domain assignments can change drastically depending on the classification scheme used, we further divided CATH-663 into two sets, depending on whether there is consensus between the definitions from CATH and ECOD (a consensus set with 313 targets and a dissensus set with 350 targets) (Fig. 2a). Based on this split, most methods perform more strongly on the consensus set, where targets may be more obvious in their domain arrangement and are easier to both classify (for CATH and ECOD) as well as predict. The opposite is true for the dissensus set, where the gap between Merizo and UniDoc widens, indicating that where CATH and ECOD disagree on an assignment, Merizo is more likely than other methods to produce a CATH-like result.

Where Merizo does not produce a well-scoring result based on CATH, the assignment may not be wrong per se, but may represent an alternative assignment that the network has negotiated given its internal knowledge of domain packing. When the possibility of an alternative ground truth is considered (such as scoring Merizo against ECOD), it can be seen that Merizo does produce ECOD-like assignments for a subset of targets despite not being trained to do so (Fig. 2b). Points on the scatterplot where a data point falls in the upper triangle indicate targets where Merizo's domain assignment matches that of ECOD over CATH, while the lower triangle represents the opposite. For some targets, the domain annotation of CATH may contain errors, or where the assignment was made from the culmination of other priors that our method does not have access to. Indeed, several cases were identified in the CATH-663 set where pairs of chains shared similar structures but were inconsistently parsed by CATH, leading to Merizo underperforming against the conflicting ground-truth labels (Supplementary Fig. 3).

When CATH or ECOD are individually used as the ground truth for scoring Merizo, it is both expected and observed that Merizo is attuned to producing CATH-like assignments (Fig. 2c). However, scoring Merizo dynamically to either CATH or ECOD (whichever ground truth scores highest), yields a much stronger performance in terms of both IoU and MCC scores, as well as the number of correctly predicted domains which increases to nearly 75%, from 65% when scoring against CATH only (Fig. 2d).

Another important facet of domain segmentation is correctly predicting the number of domains within a given target. On this task, domain count predictions by Merizo were the most accurate, scoring a mean absolute error (MAE) of 0.332 (Fig. 2e). All ML-based methods including Merizo, Eguchi-CNN and DeepDom also have a tendency to

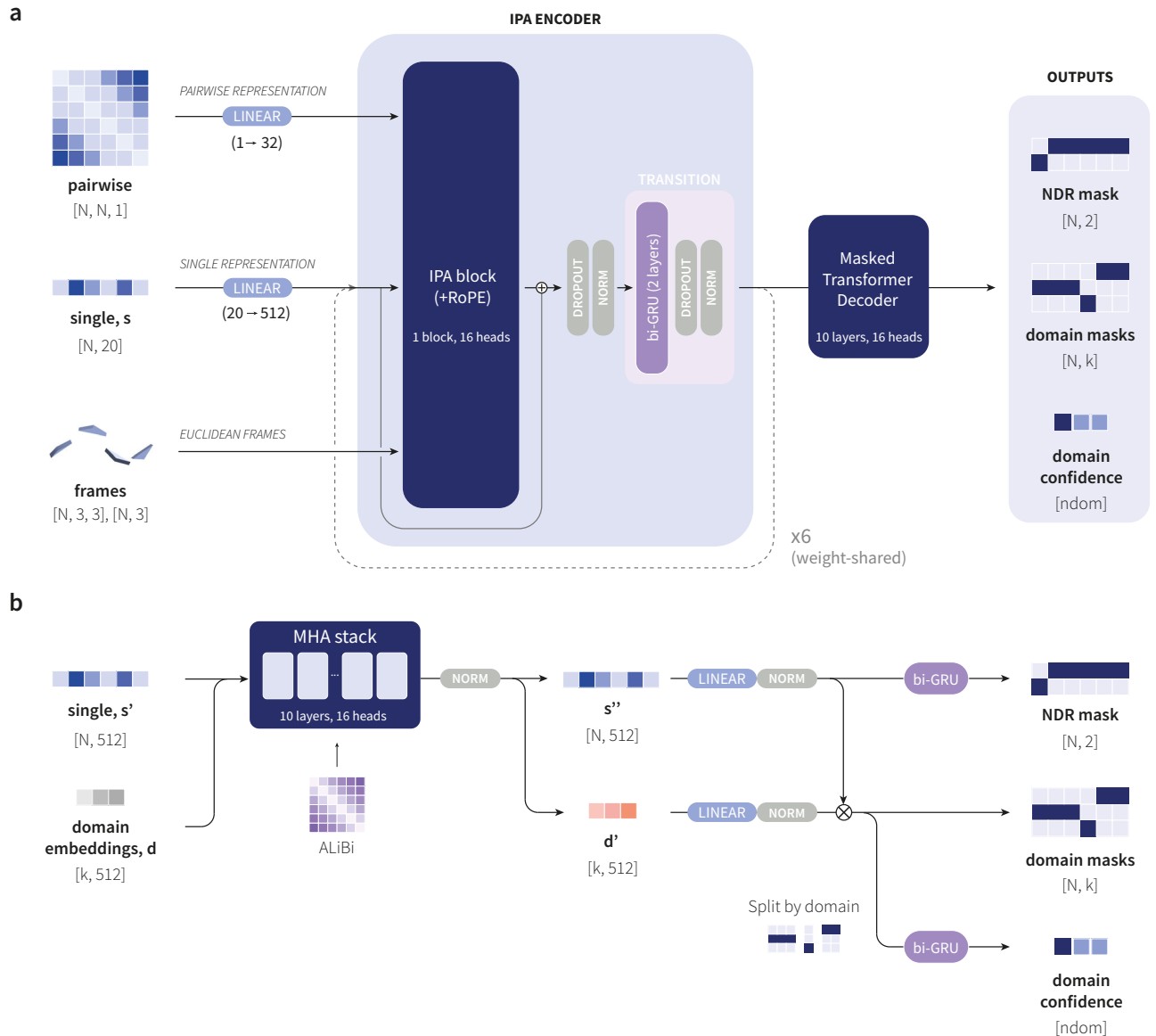

**Fig. 1 | Overview of the Merizo network. a** Summary of the network architecture. Network inputs to the IPA encoder are the single and pairwise representations and backbone frames in the style of AlphaFold2. The IPA encoder comprises six weight-shared blocks, each containing a single IPA block with RoPE positional encoding, and a bi-GRU transition block. $N$ denotes the residue count in a given target, and $k$ denotes the maximum number of assignable classes, set to 20. The encoder returns an updated single representation which is decoded by a masked transformer decoder. **b** Summary of the masked transformer decoder. In the decoder, learnable domain mask embeddings $d$ are concatenated to the single representation and passed through a 10-layer MHA stack with ALiBi positional encoding. The attention-treated output is split to recover updated single and domain mask embeddings, and each are passed through a linear layer followed by normalisation. Domain mask predictions are made via calculating the inner product between the updated single representation $s''$ and the conditioned domain embeddings $d'$. The positions of NDRs are predicted by passing $s''$ through a two-layer bi-GRU followed by projection into two dimensions. To make per-domain pIoU predictions, the predicted domain mask tensor is split according to the predicted domain and is passed through a two-layer bi-GRU, followed by projection into one dimension to produce a single pIoU value for each domain. $ndom$ represents the number of predicted domains.

underestimate rather than overestimate the number of domains, whereas the opposite is true for all other methods (Fig. 2f).

**Fine-tuning Merizo on AlphaFold2 models**
As some AFDB models may contain large stretches of unstructured regions (which we refer to as non-domain residues or NDRs), we fine-tuned Merizo on a subset of the AFDB human proteome (AFDB-human) to encourage the network to become performant on these models. Domains from AFDB-human have been classified in ECOD, describing 47,577 domains across 18,038 proteins[26]. Since Merizo is a CATH-specific domain segmentation method, we opted not to train on the ECOD classifications, but to instead utilise a self-knowledge

distillation approach which was conducted in two stages (see Methods sections 'Fine-tuning on AFDB models').

A comparison of before and after fine-tuning Merizo is shown in Fig. 3. On the task of NDR detection, we compared the number of NDRs predicted by Merizo before and after fine-tuning, as well as to three baselines: (1) predictions by UniDoc, (2) inferring from PAE/plDDT and (3) predictions by DPAM. Results show that after fine-tuning, the ability of Merizo to detect NDRs drastically increases, and is highly correlated with NDR counts inferred from PAE/plDDT and those reported by DPAM (Fig. 3a). Examples of domain assignments by Merizo before and after fine-tuning are shown in Supplementary Fig. 4. Performance on domain prediction on the CATH-663 set

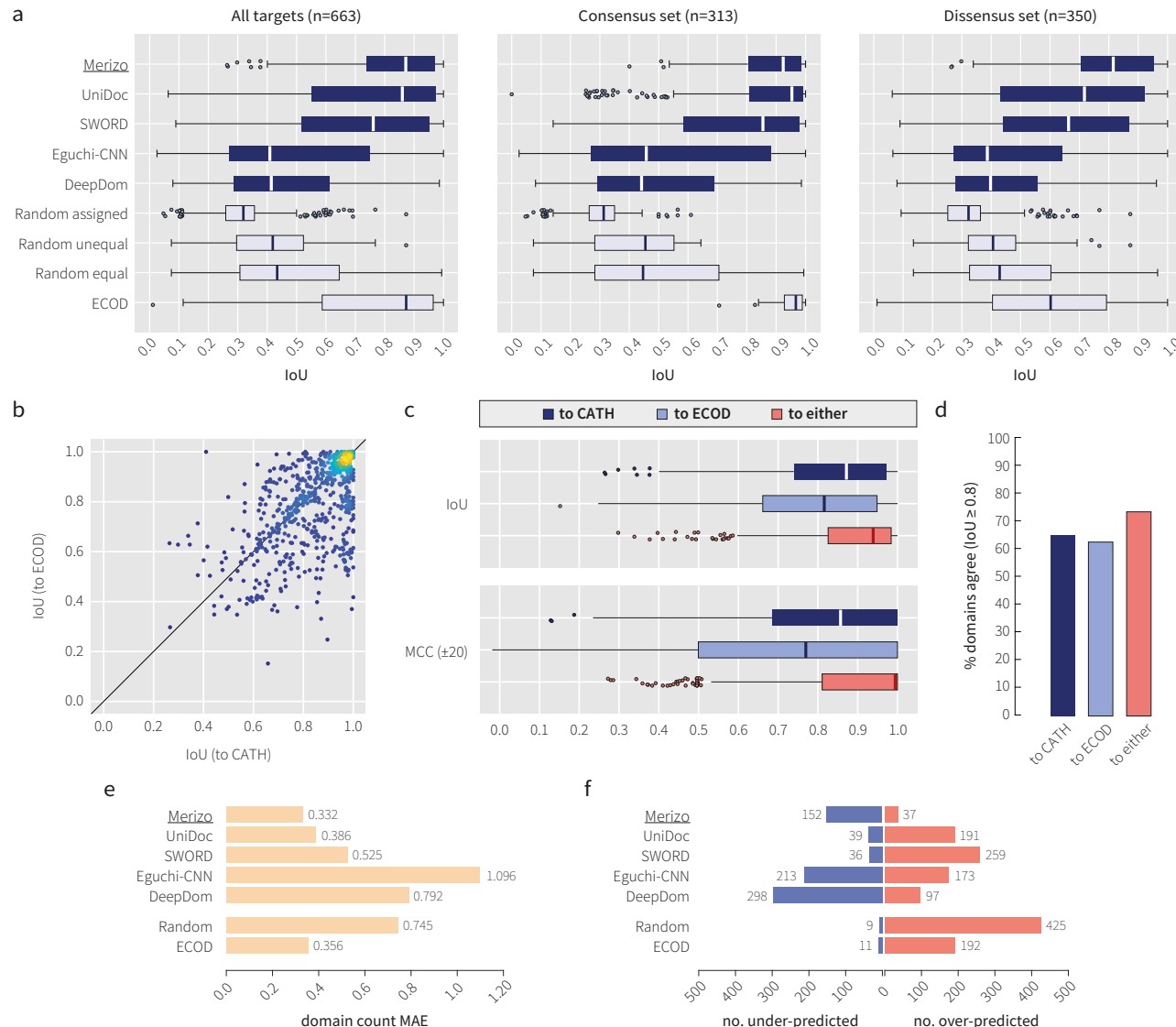

**Fig. 2 | Benchmark against existing methods on the CATH-663 set. a** IoU distributions for each method for all CATH-663 targets, and targets where there is consensus or no consensus between CATH and ECOD assignments. **b** Comparison of IoU achieved when scoring Merizo against CATH or ECOD domain assignments. The colour gradient indicates the density of data points, where yellow and dark blue are high and low respectively. **c** Box plots showing IoU and MCC (±20) distributions and **d** the percentage of domains that agree between Merizo and either CATH (dark blue), ECOD (light blue) or the maximum of either (red). Correct domains are defined as those with an IoU of at least 0.8 to the ground-truth domain. Performance of each method on **e** domain count prediction, and **f** the number of under (blue) and over-predicted (red) domains across all targets. All data shown in this figure represent the fine-tuned version of Merizo. *n* = 663 for all panels unless specified otherwise. For all box plots shown, minima and maxima are shown by the whiskers, the box limits represent the lower and upper quartiles, and solid lines inside each box represent the distribution median. Outliers are defined as data points exceeding 1.5x the interquartile range.

changes little after fine-tuning but noticeably results in a small drop in median IoU and MCC scores, but with narrower overall distributions, suggesting that self-distillation leads to more consistent assignments (Fig. 3b).

On the set of AFDB-1195, where ground-truth domain assignments are unavailable, Merizo identifies a total of 3752 domains (Fig. 3c). Despite DPAM identifying many more domains (5119) than Merizo, only 3749 (73%) were later classified into ECOD domains. One possible explanation for this discrepancy is that DPAM has detected domains that are novel to ECOD and cannot be easily assigned to an ECOD class. However, a closer examination of targets where DPAM has predicted a large number of domains revealed that DPAM had the tendency to find domains within the unstructured regions of AFDB models (Fig. 3d-e). The examples depicted in Fig. 3d, e illustrate cases where DPAM has over-segmented NDRs, leading to inflated estimates of domain counts.

Furthermore, nearly all domains (3605 domains; 96%) identified by Merizo can be aligned to the ECOD F40 representative set[3] with a TM-align score of 0.5 or greater[27] (Fig. 3c). Even at a higher threshold of 0.6, 3242 domains (86%) can be matched, suggesting that Merizo-identified domains are reasonably recognisable by ECOD (Fig. 3c).

Both SWORD and UniDoc are incapable of differentiating domains from NDRs, resulting in the inclusion of the latter in domain predictions. In models with a significant proportion of NDRs, this limitation reduces the effectiveness of both methods when applied to AFDB models, as NDRs must be addressed separately in order to accurately segment domains. We briefly explored the possibility of using a plDDT filter to remove low-quality residues, however, this commonly resulted in over-fragmented structures, unsatisfactory clean-up, or removal of residues from folded domains at higher plDDT thresholds (Supplementary Fig. 5). Assigning domains on these models using UniDoc,

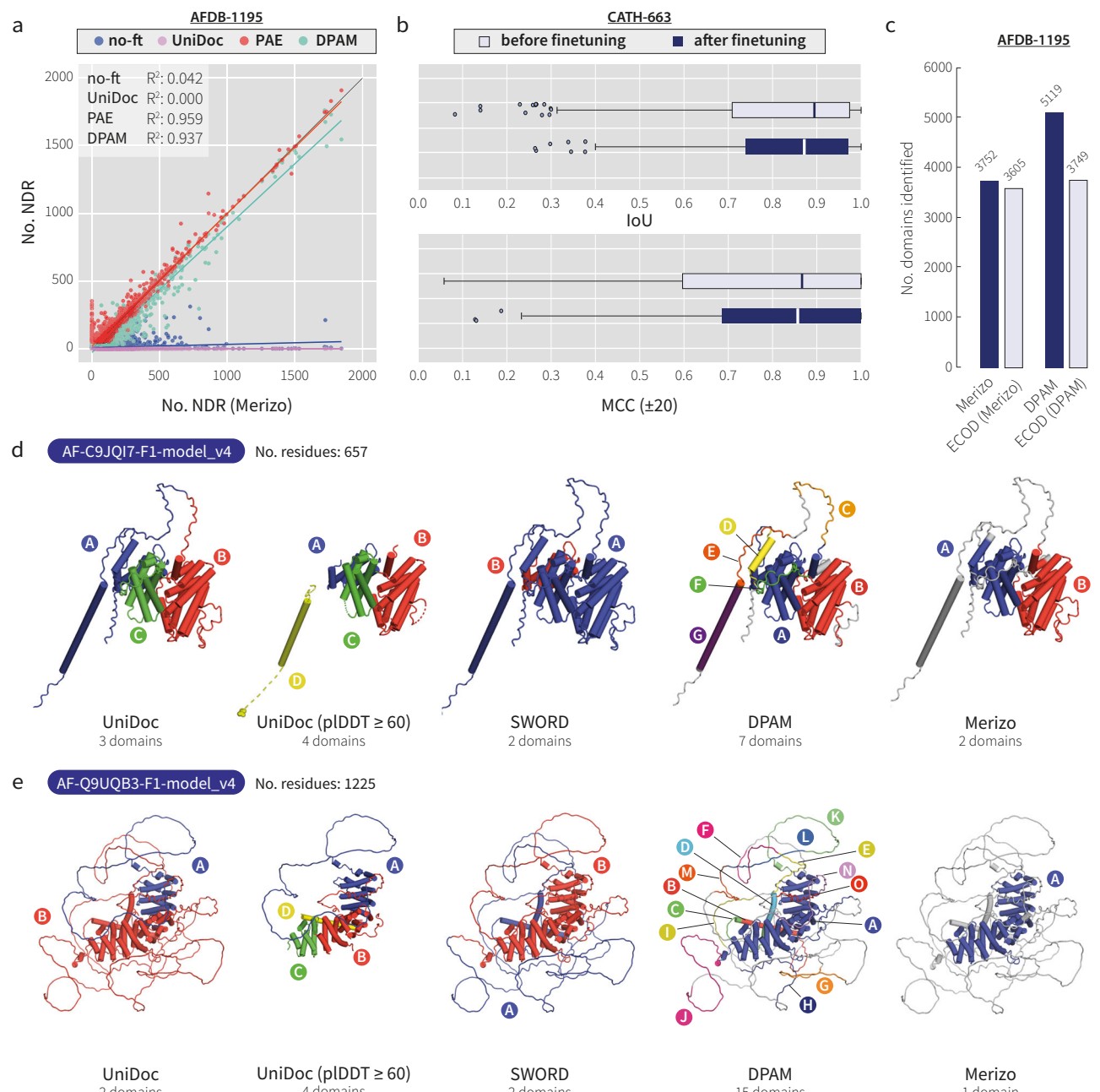

**Fig. 3 | Performance of Merizo on the AFDB-1195 set. a** Agreement between the number of NDRs identified by Merizo after fine-tuning, compared to before fine-tuning (blue), UniDoc (purple), inferring from PAE and plDDT data (red) and as classified by DPAM (teal) (n = 1195). The inset shows the coefficient of determination (R2) for each correlation. **b** IoU and MCC ( ± 20) distributions on the CATH-663 set before (light blue) and after (dark blue) fine-tuning (n = 663). For all box plots shown, minima and maxima are shown by the whiskers, the box limits represent the lower and upper quartiles, and solid lines inside each box represent the distribution median. Outliers are defined as data points exceeding 1.5x the interquartile range. **c** The number of domains identified by Merizo and DPAM (dark blue), as well as the number of identified domains matched to ECOD domains (light blue) is shown for each method. Merizo domains were assigned to ECOD representative domains via TM-align, using a threshold of 0.5 (TM-align score normalised by the length of the Merizo domain). **d-e** Examples of domain assignments by UniDoc, UniDoc (following removal of residues with plDDT less than 60), SWORD, DPAM and Merizo. Each identified domain has been shown in a different colour as well as by the text labels from A to O. NDRs are shown in white. In both examples, NDR detection is the most robust in Merizo, while UniDoc and SWORD do not classify these regions entirely, and DPAM over-segments NDRs into additional domains.

highlighted cases where UniDoc was unable to partition folded domains from leftover NDR fragments (Fig. 3d and Supplementary Fig. 5). In many cases, removing low plDDT residues altered the assignment by UniDoc (Fig. 3d and Supplementary Fig. 5). Taken together, the above observations illustrate that the presence of NDRs in AFDB models can obfuscate the performance of methods that are unable to handle such regions, leading to unsatisfactory segmentation. By employing a targeted approach whereby Merizo is fine-tuned to

recognise NDRs, domain segmentation on AFDB models can be made more robust.

## Application of Merizo to the human proteome
As a demonstration of our method, we applied Merizo to the entire AFDB-human set containing 23,391 models generated by AlphaFold2[28]. The first observation is that approximately 37% of analysed residues were classified by Merizo as NDRs (Fig. 4a). This value is supported by

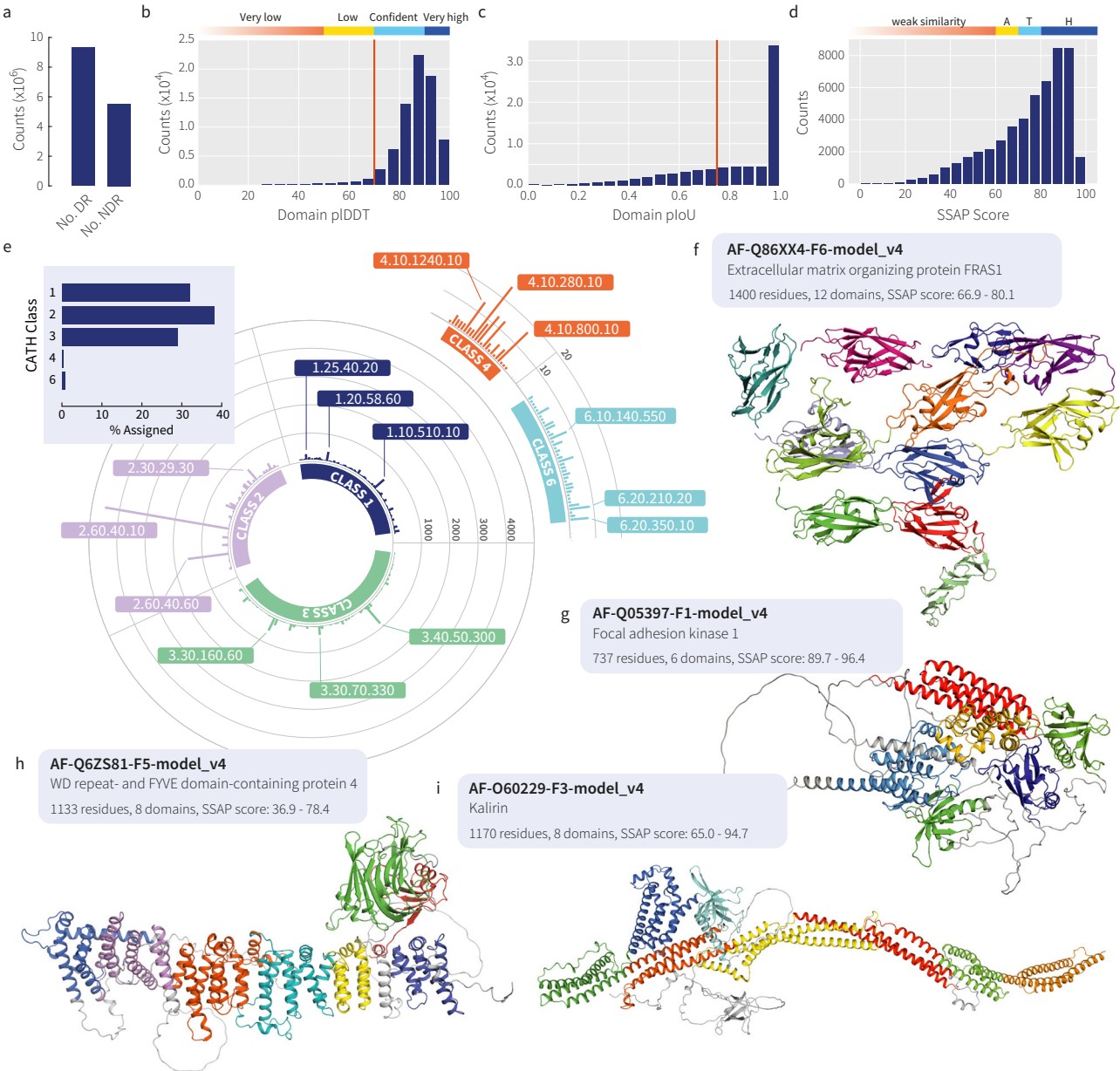

**Fig. 4 | Putative domains identified in the entire human proteome. a** The number of residues identified as domain residues (DR) and NDR in the AFDB-human set. Distributions showing the **b** average domain plDDT and **c** domain pIoU for 74,250 domains identified by Merizo in 23,391 models of the AFDB-human set. The red lines indicate thresholds applied to the plDDT and pIoU scores which were used to dichotomise a subset for further analysis. The colour bar indicates the plDDT confidence bin as per AlphaFold2, with very low (red), low (yellow), confident (light blue) and very high (dark blue) bins. **d** Distribution of SSAP scores for 50,175 confident domains, aligned to the CATH S40 non-redundant set. SSAP score bins demarcated above the histogram represent similarity at the CATH architecture (A; 70 > SSAP ≥ 60; yellow), topology (T; 80 > SSAP ≥ 70; light blue) and homologous superfamily (H; SSAP ≥ 80; dark blue) levels. SSAP scores below 60 indicate weak similarity (red). 40,818 identified domains align to CATH domains with a SSAP score of at least 60. **e** The most abundant superfamilies identified in AFDB-human by Merizo. The inset shows the distribution of domains assigned to each CATH class. **f-i** Examples of AFDB-human models segmented by Merizo, where each colour represents a different predicted domain. NDRs are shown in white.

observations from Schaeffer et al[26]. which found that globular domains comprised 62% of residues in the AFDB-human set. Overall, across 23,391 AFDB-human models, 74,250 candidate domains were identified, with most of these having high domain-level plDDT scores (residue plDDT averaged across the domain). 96.4% of these domains fall within the 'confident' to 'very high' plDDT bins demarcated by AlphaFold2[28], indicating that most domains identified by Merizo are segmented from well-folded regions where AlphaFold2 is confident (Fig. 4b).

In addition to predictions of domains, Merizo also outputs estimates of confidence in the predictions, expressed as predicted IoU

(pIoU). The pIoU distribution produced by Merizo also illustrates that the network is confident in most of its predictions, as most fall into the 0.95–1.00 pIoU bin (Fig. 4c). From our analysis of the CATH-663 set, we determined that a pIoU cut-off of 0.75 can be used to group domain predictions into high (pIoU ≥ 0.75) and low-quality (pIoU <0.75) predictions (Supplementary Fig. 6).

Next, to verify the validity of the identified domains, we extracted a subset of high-confidence domains by applying cut-offs to the domain-level plDDT (plDDT ≥ 70) and pIoU scores (pIoU ≥ 0.75). This process yielded 50,175 high-confidence domains, which were then putatively assigned to CATH superfamilies using the sequential

structure alignment program (SSAP) score[29] against the CATH S40 nonredundant set[2]. The SSAP score quantifies the similarity between two structures, taking into account the order of residues as well as secondary structure elements and motifs. SSAP scores in the ranges of 60–70, 70–80, and 80–100 correspond to similarity at roughly the architecture, topology, and homologous superfamily levels, respectively. The distribution of SSAP scores, depicted in Fig. 4d, illustrates the similarity between each high-confidence domain and the best-matching representative CATH domain. Overall, 40,818 domains (81.3%) were successfully aligned to an existing CATH class, with 49.7%, 19.2%, and 12.5% matching at the superfamily, topology, and architecture levels, respectively.

To analyse the subset of models that did not align straightforwardly to a CATH representative domain, we employed Foldseek's easy-cluster algorithm[30] for clustering these structures. Applying a criterion of 50% coverage and a TM-score threshold of 0.5, we identified 5281 distinct clusters. On closer examination, a significant number of clusters corresponded to domain fragments, including segmented blades from propeller folds and variable-sized fragments from repetitive domains like HEAT repeats. However, several clusters corresponded to structures exhibiting high domain plDDT (average residue plDDT greater than 70), which did not find a match among CATH representatives. We provide examples of these identified folds in Supplementary Fig. 7.

The superfamily distribution of the putatively assigned domains is shown in Fig. 4e. CATH classes 1 to 3 (1: mainly alpha, 2: mainly beta, and 3: alpha beta domains) comprised most of the assignments, with only roughly 1% finding matches to classes 4 and 6 (4: few secondary structures and 6: special). Like the ECOD classification of the AFDB-human set[26], the most abundant domain families included immunoglobulin-like folds such as canonical immunoglobulins (2.60.40.10) and cadherins (2.60.40.60). Domains in this topology are found in a range of proteins related to cell adhesion and immune response and are formed as long tandem repeats, including titin (Q8WZ42), sialoadhesin (Q9BZZ2), hemicentin-1 (Q96RW7), and FRAS1 (Q86XX4; Fig. 4f–i). The most abundant superfolds identified included Rossmann folds (3.40.50), jelly rolls (2.60.120), alpha/beta plaits (3.30.70), transferases (1.10.510) and helix-hairpins (1.10.287) (Supplementary Fig. 8).

### Benchmarking runtime against other segmentation methods

Besides accuracy, another important consideration of a method is its speed. Although accuracy should be the primary concern, the enormous number of models that have been made available in different databases including the AFDB, means that the speed and efficiency of methods are increasingly important factors that should not be overlooked. As such, we conducted a benchmark study of runtimes, comparing Merizo against other methods on two sets of proteins. The first is the CATH-663 benchmark set which contains proteins from 90 to 739 residues long, while the second is a small set of 27 proteins (referred to as AFDB-27) selected from the AFDB-human set, chosen to encompass the full range of lengths (up to a maximum of 2700 residues) and to test runtimes on longer models.

Table 1 shows the measured runtimes for Merizo against UniDoc, SWORD, DeepDom and Eguchi-CNN and on different hardware. For several methods including Merizo, the average runtime per target on the most optimal hardware type (CPU or GPU) is less than a fifth of a second, with the fastest method being DeepDom, which can process inputs in batches to allow them to be segmented concurrently. The slowest method in our benchmark was SWORD, which required 6366 sec (1.7 h) to process the CATH-663 set. As expected, runtimes on CPU are in general slower than on GPU hardware, however an exception to this is the UniDoc method which was 30% faster than Merizo (GPU). While UniDoc boasts faster runtime than Merizo, it is constrained by a rule whereby residues that are part of secondary

structure elements are never considered as potential domain boundaries. Although this, in theory, reduces the computational cost of the method greatly, it comes at the cost of not being able to split domains on residues which fall onto secondary structure elements. Examples of such cases can be seen in CATH, ECOD as well as in SCOPe (Supplementary Fig. 9 and Supplementary Fig. 10).

Similar results were obtained from the AFDB-27 set which compared Merizo against other high-accuracy methods (including UniDoc, SWORD and DPAM) on AFDB models. Results are shown in Supplementary Fig. 11, where it can be seen that Merizo (GPU) and UniDoc are overall the fastest methods. On targets with fewer than 1500 residues, UniDoc achieves lower runtimes than Merizo, however, the difference becomes smaller as models approach 2000 residues in length. The maximum model size that Merizo can process is limited by GPU memory; on an NVIDIA 1080Ti with 11GB of memory, this maximum is roughly 2100 residues. Models longer than this cut-off can instead be processed either on a GPU with larger memory capacity, or by CPU, albeit at an 8–10x increase in runtime. Even on a CPU, however, Merizo compares very favourably to SWORD and DPAM, which on the longer models can be up to three orders of magnitude slower than Merizo. This difference is especially prominent when comparing the total runtime of each method, as SWORD and DPAM require approximately 40 h to process the set of 27 models on a single CPU core while Merizo requires only 14 min on the same hardware. When the accuracy of each method is also taken into consideration, as well as applicability to AFDB models, the performance of Merizo compares favourably, being able to produce accurate domain assignments even on AFDB models with reasonable runtimes.

## Discussion

In this study, we have developed a fast and accurate domain segmentation method which can be applied to both experimentally-derived PDBs as well as in silico models such as those generated by AlphaFold2. AlphaFold2 models differ considerably from the former, particularly in the abundance of NDRs seen in some models (which we estimate to be around 40% of residues in the AFDB-human set). The presence of these residues can preclude some methods (including UniDoc and SWORD) from operating successfully. Despite the apparent correlation between NDRs and residues with low plDDT, it is important to note that applying a simple plDDT filter does not guarantee the successful removal of NDRs, as shown in Fig. 3d and Supplementary Fig. 5. It is conceivable that there could exist less common domains that are adequately modelled by AlphaFold2, but may be scored with lower confidence owing to their limited representation in the training data. Applying a poorly characterised plDDT filter could inadvertently lead to the exclusion of the most interesting aspects of the AFDB. The process of NDR detection is an intricate process and is addressed in Merizo by explicitly predicting their locations through a fine-tuning regime that familiarises the network with these types of residues.

Besides its speed, the major advantage of Merizo over other AFDB-centric methods such as DPAM is that Merizo requires only a PDB structure to operate, while DPAM makes use of several tools and databases (including HH-suite[31], DALI[32], Foldseek[30] and databases UniRef[30,33] PDB70[34] and ECOD[3]) as well as the PAE map from AlphaFold2. In a high-throughput setting, the minimal dependencies of Merizo make it particularly well-suited to operate on a large number of models, reducing both the amount of time spent on computation as well as on file management.

On an individual basis, one may surmise that identifying the boundaries of a domain within a single structure may not be difficult even by eye, however, the segmentation problem rapidly becomes intractable when a large number of structures are concerned. As the basic structural and functional units of protein structures, expanding our coverage of domain annotations across protein space can improve

**Table 1 | Comparison of runtimes on CATH-663 targets**

| Method | Hardware | Average time per target (s) | Total run-time (s) | Relative |
|---|---|---|---|---|
| Merizo | GPU | 0.112 | 74.32 | 1.00 |
| | CPU | 1.095 | 725.77 | 9.77 |
| UniDoc | CPU | 0.078 | 51.75 | 0.70 |
| SWORD | CPU | 9.602 | 6366.00 | 85.65 |
| DeepDom | GPU | 0.020 | 13.29 | 0.18 |
| | CPU | 0.055 | 36.69 | 0.49 |
| Eguchi-CNN | CPU | 4.475 | 2966.77 | 39.92 |

our understanding of their functions and how they interact with one another. In drug discovery, an expanded description of domains may facilitate the identification of potential targets as well as aid in repurposing existing drugs to new targets. A combination of Merizo together with structure searching or comparison tools such as Foldseek[30] or Progres[35] would be well suited for identifying the structural homologs of a protein of interest by limiting the search space to the domains that matter most.

More recently, developments in single-sequence and language model-based prediction methods[36–38] have also been accompanied by faster runtimes over traditional sequence alignment-based methods (including AlphaFold2), which will undoubtedly boost the rate at which models will be made available to the scientific community. As these methods continue to improve in predictive accuracy, it may become commonplace for predictions to be made at genome scale or above, necessitating that any downstream analysis such as domain segmentation or function prediction be prepared to process or even reprocess large amounts of data on a regular basis.

Furthermore, classification schemes such as CATH, which we used for our ground-truth labels, would benefit from having domains pre-parsed from these large model databases in order to facilitate their sorting into families. Although we have based our segmentation predictions on CATH, we recognise that other databases such as ECOD or SCOP could have been used. However, as other studies have pointed out, domain assignments for the same protein are not necessarily agreed upon between different schemes, and classification by function, secondary structure or spatial separation may give different, but equally valid assignments[16]. In the context of ML, it may be advantageous to confine the labels used for training on a single classification scheme (at least in cases where assignments by different databases are at odds with one another) in order to avoid inadvertently introducing conflicting ground truths. That being said, as shown in Supplementary Fig. 3, there are cases where Merizo has produced an assignment that matches that of ECOD but not CATH, and these cases illustrate that the network's definition of domain packing was confident enough to challenge the ground-truth CATH assignment.

Fast and accurate segmentation methods could also play a role in determining the domain arrangement of newly discovered folds and structures, which is especially applicable to exercises such as the Critical Assessment of Structure Prediction (CASP). In CASP, tools such as Merizo could be used by the organisers to determine the domain boundaries of prediction targets, particularly in the free-modelling category, in which targets have no known homologs in the Protein Data Bank. Supplementary Fig. 12 shows three multi-domain targets from the CASP15 exercise which we predicted the domain boundaries for. Two of these targets, T1170 and T1121, are annotated by the CASP organisers as two-domain proteins, however, are segmented into three plausible domains by Merizo. It is interesting to speculate how the prediction performance of some participating groups may have changed depending on the domain definitions used for assessment.

## Methods

### CATH training dataset

The PDB chains and domain annotations used for training were accessed from version 4.3 of the CATH database[2]. To later assess our method's ability to generalise to folds not seen during training, we devised a training-test split which did not overlap at the CATH homologous superfamily (H) level. Splitting the dataset at the superfamily level is imperative, as homology can occur even at low sequence identities. To generate non-overlapping training and testing datasets, we constructed an adjacency matrix containing all CATH superfamilies across classes 1 to 6. Edges were added between superfamilies if a PDB chain can be found that contains domains from two superfamilies (Supplementary Fig. 1). The resulting graph contains 655 components and is highly disproportionate, with the first and largest component containing roughly 60% of all superfamilies (2295), while the rest are spread across the other 654 components (1585 superfamilies). Additional statistics are summarised in Supplementary Table 1.

Each graph component represents a subset of PDB chains which only contain domains from an isolated set of superfamilies. Thus, by iterating over the list of components, each can be assigned to either the training or the test set without PDB chains overlapping at the H-level. As the largest component contains the majority of superfamilies and domains, it is naturally assigned to the training set. Of the remaining components, roughly 1 in 20 were held out to comprise the test set. Further redundancy filtering with CD-HIT[39] was performed to cluster targets which had a sequence identity of greater than 99%. The final training and testing set contained 17,287 and 663 chains respectively.

CATH maintains a list of ambiguous domains which have not yet been assigned to any superfamily referred to as being in the "holding pen". Such domains are unfinalized in their classification and boundary annotations. As such, they are masked out during training to avoid polluting the network by learning these regions as either single domains or NDRs.

### AFDB models used for fine-tuning

After training our network initially on the CATH dataset, Merizo was fine-tuned on models from the AFDB-human set, in order to improve predictive performance on these types of models. The AFDB-human set contains 23,391 models, however, not all models could be used for fine-tuning for several reasons. First, some AFDB-human models may contain domains that are homologous to those in the CATH-663 set and such models should be avoided. To determine a subset of AFDB-human models which did not share homologous domains with those in the CATH-663 set, we made use of ECOD domain annotations available for both datasets (standard ECOD database for CATH-663, and Schaeffer et al[26]. for AFDB-human models). Any AFDB-human model which contained an ECOD domain in the same H-group as those in the CATH-663 set were considered overlapping and thus were not suitable for training during fine-tuning. Conversely, such overlapping models were suitable for testing purposes following fine-tuning. Applying this methodology to the 18,038 AFDB-human models which had ECOD domain annotations, followed by the removal of single-domain targets and those with fewer than 200 residues (to expose the network to longer models with more varied NDRs), we were able to identify 7502 and 1195 AFDB-human models for the training and testing sets, respectively.

To determine ground-truth NDR labels for each model, we developed a proxy measure which incorporated both residue plDDT and PAE maps for each target. Residues were dichotomised into either NDR (class 0) or non-NDR (class 1) categories based on two criteria: (1) the residue plDDT is less than 60, and (2) the standard deviation of PAE values for the residue is less than 0.4. All residues meeting both criteria are assigned as NDR, and all other residues are non-NDR.

## Network architecture

Merizo is a small encoder-decoder network with approximately 37 M parameters (20.4 M in the encoder and 16.8 M in the decoder; Fig. 1). At the core of our network is the Invariant Point Attention (IPA) encoder, which makes use of the IPA module found within the structure module of AlphaFold2[20,40]. The role of the IPA module is to facilitate information mixing between the single and pairwise channels, while iteratively organising the backbone frames towards the ground-truth structure. In our usage, we repurpose the IPA module in an input-reversed fashion to instead read a folded structure into a latent representation which can then be decoded to provide the segmentation map. The IPA module takes three inputs: a single representation, pairwise representation and backbone frames. The single representation is produced by one-hot encoding the primary sequence into 20 amino acid classes and then projected into 512 feature dimensions. The dimensionality of the single representation is maintained throughout the network. For the pairwise representation, we use the pairwise distance map derived from alpha carbons, directly embedded into 32 feature dimensions as continuous values using a linear layer. Finally, the Euclidean backbone frames are calculated from each residue "frame" (N-CA-C atoms) via Gram-Schmidt orthogonalization as per AlphaFold2. Each frame consists of a rotation matrix of shape [3, 3] and translation vector of length 3.

The IPA encoder is composed of six weight-shared blocks, each with 16 attention heads and employs rotary positional encoding (RoPE[41]). In place of the typical feed-forward network that processes the attention outputs in the original Transformer model (and in AlphaFold2), we instead utilise a two-layer bi-directional gated recurrent unit (bi-GRU) which processes the post-attention single representation and introduces sequential dependency to the residue embeddings. The output of the IPA encoder is an updated single representation which is conditioned on structural information present in the pairwise and frame channels.

To make predictions from the single representation, the masked transformer decoder adapted from the Segmenter model[42] is used to predict domain masks (Fig. 1b). Learnable embeddings corresponding to $k$ domain masks, where $k$ is an arbitrary value which controls the maximum number of domains that can be assigned by the network (set to 20 in our implementation), are concatenated with the single representation (as if they were extra residues). Setting $k$ to 20 is an architectural decision and a value greater than the largest number of domains concurrently seen by the network during training is typically selected. The largest target in our training set consists of 18 domains (Supplementary Fig. 2), however with input cropping (see "Methods" section 'Training procedures'), this value will drop to approximately 2–4 domains.

The new single representation and learnable domain mask embeddings are passed through a stack of 10 multi-head attention (MHA) blocks. Each MHA block employs Attention with Linear Biases (ALiBi) style positional encoding[43] which applies a penalty to the attention score between pairs of residues, according to the separation between their residue indices. The output of the MHA stack is divided to recover the original dimensions of the single representation and learned domain mask embeddings. Per-residue domain probability distributions of shape $[N, k]$ are derived by calculating the inner product between the $[N, 512]$ single representation and $[k, 512]$ domain mask embeddings. The per-residue domain probability distributions can be converted into predicted domain masks (per-residue domain indices) via the *argmax* function.

In addition to domain masks, our network predicts two additional outputs: the positions of NDRs and per-domain pIoU predictions. NDR positions are predicted by passing the decoder-updated single representation through a two-layer bi-GRU with 256 hidden dimensions and projecting the output features of the final layer into two dimensions (with indices 0 and 1 signifying whether a residue is an NDR or not,

respectively). The argmax of the output generates a binary array of length $N$ (where $N$ is the number of residues in the input) which can be multiplied with the domain ID assignments to mask out the positions of predicted NDRs. To make pIoU predictions, the domain ID probability distributions are divided according to the predicted domain masks, and the set of residue probability distributions corresponding to each domain are passed through a two-layer bi-GRU with 512 hidden dimensions. In each case, the final timestep of the bi-GRU is projected into one dimension to predict a pIoU value between 0–1 which is trained to match the calculated IoU of the assigned domain against the ground-truth assignment.

In a final stage, the predicted domain assignments are post-processed in a cleaning step which coalesces any domain with fewer than 30 residues or any segment fewer than 10 residues, with the domain preceding it. This step is also performed on the output of the network prior to pIoU prediction by the final bi-GRU to ensure that domain pIoU is predicted on the same assignments produced by the network. During inference, an additional post-processing step is performed on the full chain input whereby if multiple domains are assigned to the same domain index by the network (which for example, can occur when a target contains a large number of domains), these domains are separated into different domains if the minimum distance between them is greater than 10 Å. This can be done by calculating the intersection between the predicted domain map and the thresholded distance map (a.k.a. contact map), as an adjacency matrix, and assigning each graph component to a new unique domain index.

## Training procedures

Our method is trained fully end-to-end in PyTorch, with all input features calculated directly from PDB files. Training was conducted in two phases: initial and fine-tuning (not to be confused with fine-tuning on AFDB models, described in Methods section 'Fine-tuning on AFDB models'). Initial training was carried out for approximately 30 epochs using the Rectified Adam (RAdam) optimiser with a learning rate of 1e-4. During training, each target chain was randomly cropped to a window of 512 residues. Fine-tuning was carried out for approximately 10 epochs in which the contribution of both the IPA and decoder loss terms were multiplied by a factor of 2. A minibatch of size 1 was used throughout, and gradients were accumulated and back-propagated every 32 mini-batches. All training was conducted using up to 6 NVIDIA GTX 1080Ti GPUs with 11GB of memory. Additional details such as the affinity learning procedure as well as loss functions are described in the Supplementary Methods section.

## Fine-tuning on AFDB models

The fine-tuning of Merizo on AFDB models was performed in two stages. First, Merizo was fine-tuned to detect NDRs in the AFDB models. NDR tuning needs to occur first before self-distillation since the predicted NDR mask overrides the domain mask. Poor performance in predicting NDRs would naturally lead to poor domain boundary prediction. As ground-truth labels for NDRs are not available, we inferred these positions via an empirically determined proxy based on residue-level PAE and plDDT scores (see Methods section 'AFDB models used for fine-tuning'). All network parameters are frozen with the exception of the bi-GRU that predicts the NDR masks. The dataset used for this exercise consists of the set of 7052 AFDB models (see Methods section 'AFDB models used for fine-tuning'). The $L_{bg,CE}$ loss component (Supplementary Methods) on the AFDB-1195 set is monitored throughout to measure network performance on the NDR task.

The second stage is conducted when the loss on the NDR task converges. All network weights are unfrozen, and training is conducted as described in Methods section 'Training procedures'. As ground-truth labels for the AFDB-human models are not available, we adopted a self-distillation approach whereby the predicted domain assignment is taken as the ground truth, following a cleaning function

that removes any domains smaller than 30 residues as well as coalesces any segments that are fewer than 10 residues with the domain preceding it. During fine-tuning, the network is trained on the 17,287 chains with CATH annotations and 7052 AFDB-human models. Network performance on the CATH-663 set is monitored to ensure that performance on this set does not degrade. Losses on AFDB models were also scaled by a factor of 0.2. Hyperparameters were the same as those described in Methods section 'Training procedures'.

## Statistics & Reproducibility

Sample sizes reported throughout the study were determined based on the availability of training and testing data available. The procedures taken to generate the training and testing split for developing our deep learning method are described clearly in the Methods section "CATH training dataset" and "AFDB models used for fine-tuning". No data were excluded from the analyses. The experiments were not randomised. The Investigators were not blinded to allocation during experiments and outcome assessment.

## Reporting summary

Further information on research design is available in the Nature Portfolio Reporting Summary linked to this article.

## Data availability

Datasets used as part of this study have been deposited to https://github.com/psipred/Merizo. Domain assignments for PDB and AFDB structures from CATH, ECOD, SCOPe and DPAM have been deposited at https://github.com/psipred/Merizo/tree/main/datasets. AlphaFold2 human proteome models used in this study can be downloaded from https://ftp.ebi.ac.uk/pub/databases/alphafold/latest/UP000005640_9606_HUMAN_v4.tar. Protein Data Bank structure files were accessed from https://www.rcsb.org including PDB 3BQC [https://doi.org/10.2210/pdb3BQC/pdb] (protein kinase CK2). Source data are provided with this paper.

## Code availability

The code and network weights of Merizo are available at https://github.com/psipred/Merizo and will be incorporated into the PSIPRED workbench at http://bioinf.cs.ucl.ac.uk/psipred/.

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

## Acknowledgements
We would like to thank Ian Sillitoe for helpful discussions on CATH domain annotations and Daniel Buchan for helpful feedback. This research was funded in whole, or in part, by the UKRI [Grant number BB/T019409/1] to D. T. J.

## Author contributions
A. M. L., S. M. K. and D. T. J. conceptualised the study. A. M. L. prepared the datasets, performed network training, and conducted data analysis and visualisation. A. M. L. and S. M. K. wrote the manuscript. All authors read and approved the final version of the manuscript.

## Competing interests
The authors declare no competing interests.
