## [Peer Review File · Nature Communications]

Merizo: a rapid and accurate protein domain segmentation method using invariant point attentionReviewer #1 (Remarks to the Author):

The manuscript introduces a novel deep-learning method for identifying protein domains. The method demonstrates superior performance compared to existing deep learning approaches, namely DeepDom and Eguchi-CNN. However, intriguingly, both deep learning methods underperform in comparison to non-machine learning methods like UniDoc and SWORD. Given that the current method was specifically trained using the CATH database, while UniDoc and SWORD were not, I am curious to know if the deep learning methods would outperform non-ML methods such as UniDoc when evaluated using ECOD or SCOP as ground truth. The authors mention that the method tends to generate CATH-like domain segmentations, hence the question arises.

Furthermore, I have two minor queries:

1. It is mentioned in the manuscript that the network was subsequently fine-tuned to handle AFDB models. It is unclear whether the benchmark results presented in Figure 2 were obtained before or after the fine-tuning process. If the results were obtained using the deep network prior to fine-tuning, I would like to know if this fine-tuning could potentially impact the network's performance on experimental structures from the PDB.

2. The fine-tuning for AFDB models utilized human proteins, while the benchmarking and applications shown in Figures 3 and 4 also involved human proteins. I am uncertain as to why models from other species, such as dogs, cats, and insects available in the AFDB, were not used for fine-tuning the network. This approach could help prevent potential biases arising from shared properties among human proteins, which might lead to an overestimation of performance.

In addition to these questions, I believe the authors have contributed a valuable tool to the scientific community. The manuscript presents a method that can accurately identify domains in both experimental structures and AlphaFold models, eliminating the need for additional dependencies like databases and sequence/structure search software. This self-contained approach enhances the usability and accessibility of the tool, making it a valuable resource for researchers.

Reviewer #2 (Remarks to the Author):

The manuscript introduces a new deep learning method, Merizo, to identify protein domains. It utilizes AlphaFold2's Invariant Point Attention (IPA) to learn embeddings of protein residues and subsequently applies clustering techniques to group these embeddings into distinct domains. The method was tested on CATH dataset and AlphaFold DB (AFDB) models. It slightly outperforms other methods on the CATH-663 dataset in terms of accuracy, but at a lower speed than UniDoc. Its advantage, as concluded by the authors, is on the AFDB models, which contain a significant proportion of non-domain residues (NDRs). I appreciate the comprehensive experiments and analysis conducted in this work. One of my major concerns is about the application/test on the AFDB models, which is explained in more details below.

Major comments:

1

The ground truth domain boundaries for the AFDB models are unknown, making it problematic to be used as a benchmark set.

2

The authors mention that "Both SWORD and UniDoc are incapable of differentiating domains from NDRs". This may be explained with caution, as both methods were not designed for such purpose. As Merizo was fine-tuned on the AFDB models to recognize NDRs, a fair comparison with SWORD and UniDoc should be based on processed AFDB models. For example, the residues with low pLDDT scores (e.g., 60 used by the authors on page 10) could be regarded as NDRs and removed before running SWORD and UniDoc. Or better way could be designed to process the AFDB models?

#3

It is less meaningful to work on predicted structure models that are mostly wrong. In my opinion, the pre-processing in the above comment is necessary for the AFDB models. The AFDB models may not contain many NDRs after pre-processing, making them suitable for further analysis, e.g., domain parsing in this work. Then they will be similar to the CATH dataset and the advantage of Merizo may disappear.

#4

The statement in the introduction that "The structure of Ephrin type-A receptor 2 (PDB 5NFK) for example, is classified in CATH as a two-domain protein (superfamilies 3.30.200.20 and 1.10.150.10), but in ECOD as a single domain (ECOD 206.1)." is incorrect.

Upon conducting a search of protein 5NFK against the CATH and ECOD databases, I verified that 5NFK is not classified as a two-domain protein in CATH, and the superfamilies 3.30.200.20 and 1.10.150.10 are not associated with 5NFK. There are no search results for 5NFK.

Additionally, my search confirmed that the superfamilies 3.30.200.20 and 1.10.150.10 do exist in CATH, but they pertain a different protein, namely 3BBT, which is distinct from 5NFK.

Regarding the ECOD classification of 5NFK, I did not find the specific ECOD ID (ECOD 206.1), associated with it. Therefore, it is unclear how this particular ECOD ID was assigned to 5NFK. It appears that there was an error in the initial description, and the provided classifications for 5NFK in both CATH and ECOD databases are inaccurate. Further investigation and clarification are required to determine the correct domain classifications for protein 5NFK in these databases.

In addition, the following explanation should be rewritten for the reasons given by CATH and ECOD for the different structural domain assignments of the 5NFK protein.

5

The description provided in section 5.3.1 of the Supplementary regarding the mapping between L_g and L_p is challenging to comprehend, particularly the statement "and obtaining the modal value of l_p predicted for residue positions where l_g is equal to $L_{g,i}$ ". The meaning of "i" is unclear, although it seems to indicate a domain index in the ground truth. The authors should provide further clarification on this matter. Moreover, the method employed to obtain the modal value of l_p is not explained, leaving it ambiguous.

The authors should elucidate how they map the predicted label to the ground truth, as the process remains unclear from the given information. Additionally, it is unclear why the maximum number of assignable classes is set to 20. It would be beneficial if the authors could provide a rationale or justification for this choice.

6

In the computation of the MCC, the parameter "m" was assessed for values of 5, 10, and 20 residues. However, only the results for the value of 20 residues are presented throughout the manuscript. What's the performance of other values?

7

For the CATH-663 test set, the ground truth is obtained from either CATH or ECOD. In order to enhance the assessment of performance, particularly for the dissensus set, I recommend the authors consider utilizing SCOPe. Unlike CATH and ECOD, which mainly rely on automatic prediction, SCOPe incorporates manual curation for many structure classifications. This manual curation process potentially yields more accurate and reliable domain assignments.

8

"Performance on domain prediction on the CATH-663 set changes little after fine-tuning but noticeably results in a small drop in median IoU and MCC scores, but with narrower overall distributions, suggesting that self-distillation leads to more consistent assignments (Figure 3b)."

I believe the IoU and MCC plots presented in Figure 3b should be inverted based on the

information shown in Figure 2c. Moreover, the plots in Figure 3b before fine tuning should match exactly with the corresponding plots in Figure 2c. However, I note that the distribution of outliers within the graph remains inconsistent and exhibits a mirrored symmetry.

Also, I think it is rather: "...after fine-tuning...results in a small increase in median IoU and MCC scores, but with wider overall distributions...". Therefore, the conclusion that "suggesting that self-distillation leads to more consistent assignments" is wrong.

9

Regarding the AFDB-1195 set, the authors reported Merizo (after fine-tuning) identified a total of 3109 domains, with about 40% of these domains matching DPAM annotations. However, only around 73% of DPAM domains were classified into ECOD domains. I am interested in how many of domains predicted by Merizo were classified into ECOD domains?

10

The authors present two examples in Figure 3d to compare the identified domains of their method with those of DPAM, UniDoc, and SWORD. The results for both examples clearly show that DPAM tends to identify domains within the unstructured regions (predicted by Merizo) of AFDB models. However, for the structure AF-C9JQ17-F1-model_v4, UniDoc, DPAM, and Merizo all successfully identify the domains in structured regions accurately. Similarly, for the structure AF-Q9UQB3-F1-model_v4, both DPAM and Merizo exhibit similar identification of the domains in structured regions.

11

In Figure 3c, only the results for Merizo after fine-tuning are presented. Considering both Figure 2b and the statement provided in the second paragraph on page 2, it is apparent that there is no notable distinction between the results of Merizo after fine-tuning and those before fine-tuning. However, the authors have reported that the implementation of a fine-tuning strategy can effectively enhance the Merizo's performance on AFDB models. Consequently, I am interested in the results achieved by Merizo before fine-tuning.

12

In the fourth paragraph on the left column of page 7, "...several clusters corresponded to well-packed domains exhibiting high regional pLDDT, which did not find a match among CATH representatives.", it is unclear how the authors determined that these clusters represent well-packed domains.

13

The authors presented four examples in Supp. Figure 7 to suggest that the assumption stating domain boundaries should not be part of the secondary structure may not be correct. The ground truth for these examples were obtained from CATH, which primarily relies on automatic prediction, while SCOPe incorporates manual curation for many structure classifications. To verify these examples, I cross-reference their domain assignments with SCOPe. Here are the SCOPe domain assignments for three out of the four examples.

(1) 1m5hA: 1-145, 146-297. The boundaries are located within coil regions, not overlapping with secondary structures.

(2) 3mfqA: 32-309. SCOPe designates it as a single-domain protein.

(3) 4dunA: 1-258. SCOPe identifies it as a single-domain protein.

However, I couldn't find any specific search results for protein 3fveA within SCOPe. Based on this analysis, it appears that the examples where domain boundaries falling on secondary structure elements lack sufficient evidence or support to be considered convincing.

14

Regarding the runtime, Merizo does not show a clear advantage, particularly compared to UniDoc. Specifically, Merizo's performance is constrained by the protein's length on the GPU or suffers a significant increase in time on the CPU, whereas UniDoc does not face such limitations.

15

In the section 4.1.2, how to access whether the AFDB-human models and the CATH-663 set are

overlapped?

16

In the last paragraph of section 4.1.2, it is unclear whether the 1195 AFDB-human models are a subset of the 7052 retained models. And how the 1195 AFDB-human models were selected?

17

In the last paragraph of section 4.2, "In a final stage, the predicted domain assignments are postprocessed in a cleaning step which coalesces any domain with fewer than 30 residues or any segment fewer than 10 residues, with the domain preceding it." How to tell the domain of less than 30 residues and a segment of less than 10 residues?

18

In the last paragraph of section 4.2, "When two domains that are distal to one another are assigned to the same domain index by the network (which for example, can occur when a target contains a large number of domains), we separate these domains if the minimum distance between them is greater than 10A."

If two domains are assigned the same domain index, indicating that all residues within these domains share the same domain index, it is unclear if they originate from distant domains. How can you tell they are from two distant domains? Do you go back and check the full-chain protein structures? If so, how do you check it? Does this mean you have to check all residues for each domain index for all proteins to see if they come from two or even more distant domains?

Minor comments:

1

I highly recommend that the authors make the necessary datasets of Merizo, which comprise the training set, test set, as well as the ground truth for both sets, publicly available.

2

In the introduction, what is "affinity learning approach"? I couldn't find a reference or more description to this method in the manuscript.

3

In section 2.1, the authors stated that they downloaded data from CATH 4.2 to train the model. However, to my knowledge, CATH 4.3 was available by at least June 2021. Therefore, I wonder why the authors did not use the updated version in the study?

4

In section 2.2, it is stated that "Merizo is a CATH-specific domain segmentation method". However, it is not clear why the authors specifically selected the CATH database as the ground truth dataset.

5

I recommend that the authors consider including the dataset name for each plot to enhance clarity. It can be challenging to refer to the dataset in the figures without this information. For instance, in Figure 3b, it would be beneficial to specify that the results are based on the CATH-663 dataset, while in Figure 3c, it should be mentioned that the prediction is related to of AFDB-1195 dataset. Without explicit dataset labels, these two plots may be mistakenly interpreted as representing the results of a single dataset.

6

The claim made in the manuscript that "For several methods including Merizo, the average runtime per target is less than a second..." is not entirely correct. Merizo does not consistently achieve an average runtime of less than one second across different hardware. It is essential to include a description of the hardware used in the study as well.

7

In the section 4.2, the authors obtained the Euclidean backbone frames for each residue using the

same algorithm in AlphaFold2. Could you explain what each dimension of [N,3,3], [N,3] represents? Other than N.

8

"...512 residues. was carried out for..." It seems that "Training" is missed before "was".

Reviewer #1 (Remarks to the Author):

1. The manuscript introduces a novel deep-learning method for identifying protein domains. The method demonstrates superior performance compared to existing deep learning approaches, namely DeepDom and Eguchi-CNN. However, intriguingly, both deep learning methods underperform in comparison to non-machine learning methods like UniDoc and SWORD. Given that the current method was specifically trained using the CATH database, while UniDoc and SWORD were not, I am curious to know if the deep learning methods would outperform non-ML methods such as UniDoc when evaluated using ECOD or SCOP as ground truth. The authors mention that the method tends to generate CATH-like domain segmentations, hence the question arises.

We would first like to thank the reviewer for their time spent on reviewing our work. This is an interesting question and we have divided our response below according to the use of ECOD or SCOP as ground truth.

Using SCOP as ground truth

When considering the performance implications of using ECOD or SCOP as the ground truth, it becomes apparent that SCOP is inherently unsuitable for serving as a benchmark in scenarios where algorithms have been specifically trained on CATH or ECOD assignments. This incompatibility arises from fundamental differences in classification methodology between SCOP and CATH. As described by Csaba et al. (reference below): *“The two hierarchies result from different protocols which may result in differing classifications of the same protein. Ignoring such differences leads to problems when being used to train or benchmark automatic structure classification methods”*.

Assignments in SCOP takes into consideration the recurrence of a fold – whether a structure has been observed to reoccur within another superfamily or if it exists independently as a single-domain (Hadley & Jones, 1999, as referenced below). These are additional priors that Merizo does not have access to when making a prediction, nor is it trained in a manner which allows it to learn this kind of evolutionary relationship. As a result, our current training regime on CATH domains would directly conflict with SCOP assignments and prevent the network from producing a SCOP-like result. Although a version of Merizo could be trained on SCOP domains, this is beyond the scope of the current study.

Using ECOD as ground truth

While ECOD’s assignment methodology also differs to that of CATH, both make assignments based on structure alone. Figure 1 (end of this document) shows the performance of each method when scored against ECOD assignments. This comparison makes use of 512 multi-domain targets from the CATH-663 set which contain at least two domains according to ECOD.

From Figure 1 (in this document), it can be seen that the performance of each method generally mirrors that of the CATH-based distributions in Figure 2a of the manuscript. Overall, the median IoU and MCC (± 20) scores obtained by Eguchi-CNN and DeepDom when scoring against ECOD, are higher than those from CATH-based scoring. However, both methods underperform compared to Merizo and both non-ML UniDoc and SWORD. Given that all ML-based methods (Merizo, Eguchi-CNN and DeepDom) were trained on CATH annotations and Merizo achieves high scores in both CATH and ECOD-based settings, it does not appear that poor performance by Eguchi-CNN and DeepDom are the result of ground truth misalignment. That being the case, lower performance by

DeepDom is expected, given that the method only uses sequence information, however in the case of Eguchi-CNN (which uses structural information), poor performance may be attributed to the lack of a homology-based training and test split, leading to overestimation of performance during training. Figure 1b (in this document) showcases several examples where Eguchi-CNN produces noisy predictions where individual residues are assigned to individual domains. In these cases, poor predictive performance stems from a lack of generalisation from the neural network and not from an alternative assignment being predicted.

We hope that the above addresses the reviewer's query regarding the performance of ML and non-ML methods when an alternative ground truth is considered.

References:

Csaba, G., Birzele, F. and Zimmer, R., 2009. Systematic comparison of SCOP and CATH: a new gold standard for protein structure analysis. *BMC structural biology*, 9, pp.1-11.

Hadley, C. and Jones, D.T., 1999. A systematic comparison of protein structure classifications: SCOP, CATH and FSSP. *Structure*, 7(9), pp.1099-1112.

Furthermore, I have two minor queries:

2. It is mentioned in the manuscript that the network was subsequently fine-tuned to handle AFDB models. It is unclear whether the benchmark results presented in Figure 2 were obtained before or after the fine-tuning process. If the results were obtained using the deep network prior to fine-tuning, I would like to know if this fine-tuning could potentially impact the network's performance on experimental structures from the PDB.

We apologise for the confusion. The benchmarking shown in Figure 2 of the manuscript (on the CATH-663 set) shows the performance of Merizo after fine-tuning on AFDB NDRs. Results on performance before and after fine-tuning is only shown in Figure 3a-b of our submission. We have clarified this in the figure legend.

3. The fine-tuning for AFDB models utilized human proteins, while the benchmarking and applications shown in Figures 3 and 4 also involved human proteins. I am uncertain as to why models from other species, such as dogs, cats, and insects available in the AFDB, were not used for fine-tuning the network. This approach could help prevent potential biases arising from shared properties among human proteins, which might lead to an overestimation of performance.

We thank the reviewer for this interesting question. In truth, any subset of the AFDB could have been utilized, provided it contained sufficient targets with NDRs to learn from. We were however constrained by the fact that determining a subset of AFDB models which did not overlap our CATH-663 set (in terms of domain homology), required ground truth assignments, which were sourced from ECOD. These assignments were unfortunately only available for the AFDB human set (Schaeffer et al., reference below). We have added additional details in the amended Methods section 'AFDB models used for fine-tuning' which describes the dataset selection procedure.

Regarding the reviewer's comment about mitigating bias learned from human proteins – during fine-tuning, losses are only backpropagated on the NDR residues (as well as predicted assignments

that are unequivocally wrong, for example, domains containing single residues), which minimises the likelihood that the network learns any information from the non-NDR regions of human proteins specifically, other than correcting small errors. We hope this addresses the reviewer's concern.

References:

Schaeffer, R.D., Zhang, J., Kinch, L.N., Pei, J., Cong, Q. and Grishin, N.V., 2023. Classification of domains in predicted structures of the human proteome. *Proceedings of the National Academy of Sciences*, 120(12), p.e2214069120.

In addition to these questions, I believe the authors have contributed a valuable tool to the scientific community. The manuscript presents a method that can accurately identify domains in both experimental structures and AlphaFold models, eliminating the need for additional dependencies like databases and sequence/structure search software. This self-contained approach enhances the usability and accessibility of the tool, making it a valuable resource for researchers.

We sincerely thank the reviewer for investing their time in reviewing our paper.

Reviewer #2 (Remarks to the Author):

The manuscript introduces a new deep learning method, Merizo, to identify protein domains. It utilizes AlphaFold2's Invariant Point Attention (IPA) to learn embeddings of protein residues and subsequently applies clustering techniques to group these embeddings into distinct domains. The method was tested on CATH dataset and AlphaFold DB (AFDB) models. It slightly outperforms other methods on the CATH-663 dataset in terms of accuracy, but at a lower speed than UniDoc. Its advantage, as concluded by the authors, is on the AFDB models, which contain a significant proportion of non-domain residues (NDRs). I appreciate the comprehensive experiments and analysis conducted in this work. One of my major concerns is about the application/test on the AFDB models, which is explained in more details below.

We would like to first thank the reviewer for dedicating time to evaluate our manuscript.

Major comments:

1. The ground truth domain boundaries for the AFDB models are unknown, making it problematic to be used as a benchmark set.

We apologise if we gave the impression that the AFDB models were used for benchmarking. The only benchmarking performed in our study was on the CATH-663 set in terms of CATH ground truths (Results section 'Benchmark against existing state-of-the-art methods' and Figure 2 of the manuscript) and runtime benchmarks (Results section 'Benchmarking the runtime of Merizo against other segmentation methods' and Table 1). For the reason that the reviewer has mentioned, we do not conduct any benchmarking on AFDB models, bar several qualitative comparisons based on NDR handling in Figure 3d of the manuscript.

2. The authors mention that "Both SWORD and UniDoc are incapable of differentiating domains from NDRs". This may be explained with caution, as both methods were not designed for such purpose. As Merizo was fine-tuned on the AFDB models to recognize NDRs, a fair comparison with SWORD and UniDoc should be based on processed AFDB models. For example, the residues with low pLDDT scores (e.g., 60 used by the authors on page 10) could be regarded as NDRs and removed before running SWORD and UniDoc. Or better way could be designed to process the AFDB models?

We appreciate the reviewer's feedback and have taken their suggestions into careful consideration. One of the key strengths of our Merizo method, as highlighted in our manuscript, is its applicability to both experimental and AFDB models. Notably, in the case of the latter, NDRs are handled without requiring additional processing from users. We recognise that the design of SWORD and UniDoc may not align with this functionality, but the underlying distinction remains a valid one.

Unfortunately, the process of removing NDRs is not as straightforward as removing low-pLDDT residues as suggested by the reviewer. To illustrate this point, we have introduced a new Supplementary Figure 5 (Figure 2 below) that provides representative examples of UniDoc assignments before and after the removal of residues with pLDDT values below 60. Additionally, we have extended Figure 3d of the manuscript to include new panels showcasing the impact of this

filtering approach on the performance of UniDoc. Within most structures containing NDRs, the elimination of residues guided solely by pLDDT leads to a myriad of irregularities, and introduces new problems to the structures.

Figure 2a in this document showcases scenarios where pLDDT-based filtering inadequately addresses NDRs, leading to incomplete clean-up. In these cases, UniDoc incorrectly assigns fragments into the same domain as folded regions, as the method has no ability to ignore these fragments. These fragments range from single residues to long stretches of NDRs which cannot be reliably removed even when a more stringent pLDDT threshold of 70 is employed. Using higher pLDDT thresholds also carries the risk of inadvertently eliminating residues from folded regions (Figure 2a-iii and 2b).

All of the above complexities highlight that NDR detection is non-trivial. While sometimes apparent to the human eye, automating their removal demands nuanced approaches. This is especially important given that it is estimated by both our study and that of Schaeffer et al. (reference below), that NDRs comprise an estimated 40% of residues in the AFDB human proteome. This is why, for our fine-tuning exercise, we developed a composite metric where we combined both pLDDT as well as PAE information to identify NDR residues, but this is also an imperfect measure and we capitalised on the intrinsic averaging nature of deep learning to make sense of imperfect signals.

Furthermore, the lack of a comprehensive study of domain compositions within AFDB hinders our understanding of potential correlations between AlphaFold2's confidence scores (pLDDT and pTM) and certain domain types. It is conceivable that less common domains might be adequately modelled by AlphaFold2 but are scored with lower confidence due to their limited representation in training data. Introducing a poorly characterised filter to such models could inadvertently exclude the most interesting aspects of the AFDB.

With this in mind, we hope the reviewer will agree that there is not really a “fair” comparison that can be done as we also cannot impose a crude NDR removal method onto SWORD and UniDoc for comparison to Merizo, nor is the scope of our manuscript to address the functional gaps of other methods. We would certainly welcome the authors of these programs to consider incorporating NDR handling into their methods to allow for a more balanced comparison in the future.

References:

Schaeffer, R.D., Zhang, J., Kinch, L.N., Pei, J., Cong, Q. and Grishin, N.V., 2023. Classification of domains in predicted structures of the human proteome. *Proceedings of the National Academy of Sciences*, 120(12), p.e2214069120.

3. It is less meaningful to work on predicted structure models that are mostly wrong. In my opinion, the pre-processing in the above comment is necessary for the AFDB models. The AFDB models may not contain many NDRs after pre-processing, making them suitable for further analysis, e.g., domain parsing in this work. Then they will be similar to the CATH dataset and the advantage of Merizo may disappear.

As discussed in the previous point, the removal of NDRs is a nuanced and complex exercise and is precisely the advantage that currently distinguishes Merizo from alternative methods.

While the reviewer's suggestion to pre-process the AFDB models is welcome, it is crucial to highlight that this procedure does not yield satisfactory clean-up of NDRs, but on the contrary, introduces

new structural irregularities that methods such as UniDoc are unable to navigate effectively. Presenting the outcomes of SWORD and UniDoc, following a simplistic NDR removal process, alongside the specialised approach of Merizo would yield a misleading comparison that does not reflect performance as would be experienced by users. Furthermore, even setting aside the discussion of NDR removal, without ground-truth assignments, we would also be limited to making qualitative comparisons between methods on individual cases.

For these reasons, we believe that each method should be evaluated on its current merits, so as to maintain focus on the outcomes of our study. The presence of NDRs in AFDB models and the extra complexities that they bring cannot be overlooked, and we believe that potential users should be well-informed about the performance characteristics of each method.

4. The statement in the introduction that “The structure of Ephrin type-A receptor 2 (PDB 5NFK) for example, is classified in CATH as a two-domain protein (superfamilies 3.30.200.20 and 1.10.150.10), but in ECOD as a single domain (ECOD 206.1).” is incorrect.

Upon conducting a search of protein 5NFK against the CATH and ECOD databases, I verified that 5NFK is not classified as a two-domain protein in CATH, and the superfamilies 3.30.200.20 and 1.10.150.10 are not associated with 5NFK. There are no search results for 5NFK.

Additionally, my search confirmed that the superfamilies 3.30.200.20 and 1.10.150.10 do exist in CATH, but they pertain a different protein, namely 3BBT, which is distinct from 5NFK.

Regarding the ECOD classification of 5NFK, I did not find the specific ECOD ID (ECOD 206.1), associated with it. Therefore, it is unclear how this particular ECOD ID was assigned to 5NFK. It appears that there was an error in the initial description, and the provided classifications for 5NFK in both CATH and ECOD databases are inaccurate. Further investigation and clarification are required to determine the correct domain classifications for protein 5NFK in these databases.

In addition, the following explanation should be rewritten for the reasons given by CATH and ECOD for the different structural domain assignments of the 5NFK protein.

The reviewer is correct, and we apologise for the oversight. The wrong protein and PDB were referenced in our original submission, and we have now corrected this to read:

“The structure of protein kinase CK2 (PDB 3BQC chain A) for example, is classified in CATH as a two-domain protein (superfamilies 3.30.200.20 and 1.10.510.10), but in ECOD as a single domain (ECOD 206.1.1.24) (Schaeffer et al., 2021).”

5. The description provided in section 5.3.1 of the Supplementary regarding the mapping between Lg and Lp is challenging to comprehend, particularly the statement “and obtaining the modal value of Lp predicted for residue positions where Lg is equal to Lg,i”. The meaning of “i” is unclear, although it seems to indicate a domain index in the ground truth. The authors should provide further clarification on this matter. Moreover, the method employed to obtain the modal value of Lp is not explained, leaving it ambiguous.

The authors should elucidate how they map the predicted label to the ground truth, as the process remains unclear from the given information.

We apologise that this was unclear in our submission. We have now re-written Supplementary Methods section ‘Domain pairing’ and included Algorithm 1 which describes the step-by-step process of domain pairing.

To answer the reviewer’s comment more specifically, regarding how the modal value of l_p is determined – by “modal value” we refer to the mode, i.e. most common value in the l_p array at the residue positions of the domain being evaluated. For example, when evaluating ground truth domain 1, determining the most common non-zero value within the region of l_p below in boldface (which would be index 7):

$l_g = [\mathbf{1}, \mathbf{1}, \mathbf{1}, \mathbf{1}, \mathbf{1}, 2, 2, 2, 2, 2]$ (ground-truth domain assignment)

$l_p = [\mathbf{0}, \mathbf{0}, \mathbf{7}, \mathbf{7}, \mathbf{7}, \mathbf{7}, 4, 4, 4, 4]$ (predicted domain assignment)

In this example, ground-truth domain 1 would be paired with predicted domain index 7, and ground truth domain 2 would be paired with predicted domain index 4.

Note that the above uses notation from our first submission, while the amended text uses new notation where domain indices are represented by k , to be consistent with Supplementary Method section ‘Learning to cluster residues via embedding affinity’.

Additionally, it is unclear why the maximum number of assignable classes is set to 20. It would be beneficial if the authors could provide a rationale or justification for this choice.

The maximum number of assignable classes is an arbitrary value and an architectural hyperparameter, and is set to be greater than the maximum number of domains in the training set (18 domains). We have expanded this explanation in paragraph 3 of Methods section ‘Network architecture’.

6. In the computation of the MCC, the parameter “m” was assessed for values of 5, 10, and 20 residues. However, only the results for the value of 20 residues are presented throughout the manuscript. What's the performance of other values?

We apologise for this oversight. Originally, the manuscript showed results at all three levels of m , however conveyed redundant information at the additional cost of extra complexity and redundant figure panels. As such, we opted to show results for $m=20$ as this best shows which methods best capture the ground truth assignments without over-penalising boundary precision. As we do not show data from $m=5$ or 10, we have removed these from the description of the methods, as well as expanded the justification in Supplementary Methods section ‘Matthew’s Correlation Coefficient’.

7. For the CATH-663 test set, the ground truth is obtained from either CATH or ECOD. In order to enhance the assessment of performance, particularly for the dissensus set, I recommend the authors consider utilizing SCOPe. Unlike CATH and ECOD, which mainly rely on automatic prediction, SCOPe incorporates manual curation for many structure classifications. This manual curation process potentially yields more accurate and reliable domain assignments.

This is an interesting question asked by both reviewers and we refer the reviewer to the response provided for point 1 of reviewer 1.

In response to the reviewer, it appears that the reviewer is mistaken regarding the interchangeability of domain classification databases. While we have used both CATH and ECOD classifications as ground truths in our study, we wish to emphasise that there are fundamental but unignorable differences in the assignment strategies between CATH and SCOPe. SCOPe assignments are informed by fold recurrence while those of both CATH and ECOD are not. As Merizo does not have access to such information (nor has been trained to make use of it), the method would not be expected to perform well under these standards. As explained in our response to reviewer 1, a method trained on annotations not incorporating fold recurrence information would directly be at odds with SCOPe assignments. For this reason, it would be inadvisable to assess a method trained on CATH or ECOD assignments against those of SCOPe. This could be remedied by training an entirely new Merizo based on SCOPe assignments, which would then invalidate comparisons against CATH-like predictors such as those benchmarked against in this study.

Respectfully, we hold a differing opinion from the notion that the assignments of SCOPe are “more accurate” than those of CATH, as the two rely on entirely different methodologies for domain assignment. We would like to clarify that our study's objective is not to ascertain a definitive “correct” domain assignment, but rather to evaluate each method's performance in generating assignments akin to CATH, which Merizo was trained to produce. This is why throughout the manuscript we have used language such as “CATH-like” to describe the performance of Merizo against other methods. We apologise if this was unclear from our initial manuscript and we hope that the above discussion has clarified this message.

8. "Performance on domain prediction on the CATH-663 set changes little after fine-tuning but noticeably results in a small drop in median IoU and MCC scores, but with narrower overall distributions, suggesting that self-distillation leads to more consistent assignments (Figure 3b)."

I believe the IoU and MCC plots presented in Figure 3b should be inverted based on the information shown in Figure 2c. Moreover, the plots in Figure 3b before fine tuning should match exactly with the corresponding plots in Figure 2c. However, I note that the distribution of outliers within the graph remains inconsistent and exhibits a mirrored symmetry.

Also, I think it is rather: "...after fine-tuning...results in a small increase in median IoU and MCC scores, but with wider overall distributions...". Therefore, the conclusion that "suggesting that self-distillation leads to more consistent assignments" is wrong.

Upon investigation, it appears that the legend was incorrect in Figure 3b of the manuscript and has now been fixed. We would like to note that the data shown in Figure 2 of the manuscript also represents the fine-tuned network and the ‘after fine-tuning’ box in Figure 3b now matches the distribution shown in Figure 2a and 2c. Throughout the paper, unless specified, we show data for the fine-tuned network.

The original sentence “Performance on domain prediction on the CATH-663 set changes little after fine-tuning but noticeably results in a small drop in median IoU and MCC scores, but with narrower

overall distributions, suggesting that self-distillation leads to more consistent assignments”, is correct following amendment.

9. Regarding the AFDB-1195 set, the authors reported Merizo (after fine-tuning) identified a total of 3109 domains, with about 40% of these domains matching DPAM annotations. However, only around 73% of DPAM domains were classified into ECOD domains. I am interested in how many of domains predicted by Merizo were classified into ECOD domains?

This is an interesting question. To provide an answer to this comment, we conducted a more thorough analysis on the number of Merizo-identified domains that could be matched to ECOD. We refer the reviewer to the new Figure 3c in the revised manuscript.

The first change to notice is that the number of domains identified by Merizo has been amended to 3752 (from 3109 in the previous submission). The reason for this discrepancy is that the previous value used an incorrect ploU threshold which has now been amended to 0.75 in line with the rest of the manuscript.

We have further removed the ‘common’ entry as we feel this complicates the new analysis, given that commonality was previously determined via overlap between the predicted residue ranges, and not structural alignment, as has been performed in the new analysis. We feel that presenting data conveying similarity, where different measures of similarity were used, would be confusing for the reader.

To determine the number of domains that can be matched to ECOD domains, we aligned each putative Merizo-identified domain to the ECOD F40 representative set using TM-align. Of the 3752 domains, 3605 domains (96%) could be matched to a domain in ECOD based on a TM-align threshold of 0.5, indicating that the two structures are likely to share the same fold (Xu & Zhang, reference below). Even at a higher TM-align score threshold of 0.6, 3242 (86%) domains could be matched to ECOD, demonstrating that most identified domains represent reasonably recognisable structures segmented from the AFDB models.

References:

Xu, J. and Zhang, Y., 2010. How significant is a protein structure similarity with TM-score= 0.5?. *Bioinformatics*, 26(7), pp.889-895.

10. The authors present two examples in Figure 3d to compare the identified domains of their method with those of DPAM, UniDoc, and SWORD. The results for both examples clearly show that DPAM tends to identify domains within the unstructured regions (predicted by Merizo) of AFDB models. However, for the structure AF-C9JQ17-F1-model_v4, UniDoc, DPAM, and Merizo all successfully identify the domains in structured regions accurately. Similarly, for the structure AF-Q9UQB3-F1-model_v4, both DPAM and Merizo exhibit similar identification of the domains in structured regions.

The reviewer’s comment regarding the similarity of assignments by Merizo and DPAM in the structured regions of the two examples is valid, however ignores the fact that Merizo is the only method shown that can ignore the long strings of NDRs in each example. The over-assignment problem by DPAM is despite the method being tailored for AFDB models. The advantage of Merizo is especially apparent when the runtime of DPAM is considered (Supplementary Figure 11). As

explained in our response to point 2, NDR removal is non-trivial and Merizo's ability to ignore these residues when making an assignment should not be overlooked. The fact remains that without additional and effective processing of NDRs, the direct usability of methods such as DPAM, UniDoc and SWORD on AFDB models by many users, including non-specialists, will be limited.

11. In Figure 3c, only the results for Merizo after fine-tuning are presented. Considering both Figure 2b and the statement provided in the second paragraph on page 2, it is apparent that there is no notable distinction between the results of Merizo after fine-tuning and those before fine-tuning. However, the authors have reported that the implementation of a fine-tuning strategy can effectively enhance the Merizo's performance on AFDB models. Consequently, I am interested in the results achieved by Merizo before fine-tuning.

Our statement of "[fine-tuning] can effectively improve the performance of Merizo on AFDB models" refers to the ability of Merizo to differentiate between domain and non-domain residues, since only after fine-tuning can NDRs be detected. As shown in Figure 3a, NDR detection by Merizo is virtually non-existent prior to fine-tuning. We apologise that this was not made clear in our original submission, and we have amended the final line of Results section 'Fine-tuning Merizo on AlphaFold2 models' to refer to NDR detection more explicitly.

Furthermore, we have included a new Supplementary Figure 4 (Figure 3 below), where we show and compare the segmentation results of Merizo before and after fine-tuning was conducted. As seen in the figure, prior to fine-tuning, domains were regularly identified in NDR regions, which do not occur after our fine-tuning exercise. We hope that this addresses the reviewer's query.

12. In the fourth paragraph on the left column of page 7, "...several clusters corresponded to well-packed domains exhibiting high regional pLDDT, which did not find a match among CATH representatives.", it is unclear how the authors determined that these clusters represent well-packed domains.

The compactness of the domains shown in Supplementary Figure 7 (previously Supplementary Figure 5) were determined through visual inspection. We understand that this statement may be misleading and so we have amended the main text to now read:

"However, several clusters corresponded to structures exhibiting high domain pLDDT, which did not find a match among CATH representatives".

In Supplementary Figure 7 and its legend, we have amended it to include the minimum domain pLDDT of these structures, and the text to now read:

"The cluster size as well as the lowest domain pLDDT (average pLDDT across the domain) is shown for each example."

Furthermore, we have identified two errors in the quoted cluster sizes shown in the original submission. This concerns panels a and f of Supplementary Figure 7.

13. The authors presented four examples in Supp. Figure 7 to suggest that the assumption stating domain boundaries should not be part of the secondary structure may not be correct. The ground truth for these examples was obtained from CATH, which primarily relies on automatic prediction,

while SCOPe incorporates manual curation for many structure classifications. To verify these examples, I cross-reference their domain assignments with SCOPe. Here are the SCOPe domain assignments for three out of the four examples.

(1) 1m5hA: 1-145, 146-297. The boundaries are located within coil regions, not overlapping with secondary structures.

(2) 3mfqA: 32-309. SCOPe designates it as a single-domain protein.

(3) 4dunA: 1-258. SCOPe identifies it as a single-domain protein.

However, I couldn't find any specific search results for protein 3fveA within SCOPe. Based on this analysis, it appears that the examples where domain boundaries falling on secondary structure elements lack sufficient evidence or support to be considered convincing.

To our knowledge, there is no requirement that domain boundaries must not be part of secondary structure elements (SSE). George & Heringa (reference below), conducted a comprehensive study of protein domain linkers, finding an abundance of SSEs, including that small linkers (approximately 4-5 residues in length) primarily adopted β -strands. The proportion of residues in non-coil conformations in linkers of all sizes were not insignificant (43% and 31% helical in medium and large linkers).

Prompted by the reviewer's question, we conducted an analysis of SCOPe domain classifications for which we identified 45,693 PDB chains annotated by SCOPe as having at least two domains. Of this number, we were able to find 1855 cases where, based on secondary structure classification using STRIDE (reference below), the residues immediately preceding or following the specified SCOPe boundary were characterised by a non-coil secondary structure. Specifically, we observed 494 cases of domain boundaries being found in alpha helices and 304 in β -strands.

Several examples are shown in a new Supplementary Figure 10 (Figure 4 below). In panel a), three examples of structures where domain boundaries assigned by SCOPe fall onto an alpha helix. In two cases (panels i and iii), the parse given by Merizo predicts the boundary to be at the end of the SSE, while UniDoc predicts both structures as single domains, perhaps due to a missed cut point given its SSE constraint.

Furthermore, we show an example of calmodulin in panel b), which is formed of two roughly symmetrical EF-hand domains. In holo-calmodulin, the domain linker adopts a helical structure. SCOPe classifies this structure of calmodulin as a single domain protein. In contrast, Merizo segments the structure directly at the centre of the central helix, preserving the symmetry of the two halves, while UniDoc assigns two asymmetric domains due to the SSE constraint.

Even ignoring the above discussion, it is worth noting that UniDoc was evaluated by its authors, in part, against CATH, wherein domain boundaries do occasionally fall on secondary structure elements. Based on this, we maintain the perspective that unless there exists compelling evidence that domain boundaries unequivocally cannot fall on secondary structure elements, the imposition of such an arbitrary rule on a method appears unjustified.

Moreover, it appears that one of the contributing factors to the speed of UniDoc's execution is precisely due to the SSE-exclusion rule. By not considering any residues part of SSEs as boundary points, the computational cost of the method is significantly lessened, but as we show, at the expense of accuracy, as some boundaries in CATH, ECOD and even SCOPe, do indeed fall on SSEs.

We hope that the points discussed above as well as the new analysis conducted on SCOPe domain boundaries, sufficiently addresses the reviewer's point.

References:

George, R.A. and Heringa, J., 2002. An analysis of protein domain linkers: their classification and role in protein folding. *Protein engineering*, 15(11), pp.871-879.

Heinig, M. and Frishman, D., 2004. STRIDE: a web server for secondary structure assignment from known atomic coordinates of proteins. *Nucleic acids research*, 32(suppl_2), pp.W500-W502.

14. Regarding the runtime, Merizo does not show a clear advantage, particularly compared to UniDoc. Specifically, Merizo's performance is constrained by the protein's length on the GPU or suffers a significant increase in time on the CPU, whereas UniDoc does not face such limitations.

While UniDoc certainly boasts superior speed and cost efficiency compared to Merizo, we hold a respectful difference of opinion with the reviewer regarding Merizo's purported lack of a clear advantage over UniDoc. It is important to recognize that Merizo presents a range of attributes not found UniDoc, such as its ability to predict NDRs, its alignment to CATH (Figure 2c of the manuscript), and the absence of an SSE constraint for domain boundaries.

As previously discussed in point 13, the speed of UniDoc may increase significantly if residues part of SSEs were to be considered as cut points. Nevertheless, we feel that every method should be evaluated based on their current merits, and UniDoc in its current form is not able to work optimally on AFDB models with many NDRs. We believe that this is an important distinction between Merizo and UniDoc and should not be overlooked.

The reviewer's comment regarding Merizo's length constraint is valid, however we apologise that we have not made clear that this constraint is dependent on the GPU used for computation and will vary depending on what hardware users have access to. The GTX 1080 Ti GPU used for our testing possesses 11GB of memory, which is widely regarded as insufficient for most modern deep learning applications. Running Merizo on more up-to-date hardware will remove the length ceiling as well as potentially decrease runtime. In the future, we aim to improve the Merizo method to minimise its memory footprint and to decrease runtime to allow effective inference on a wider range of hardware. We thank the reviewer for their thoughts and comments which we believe motivates us to concentrate our methods on improving the aspects which matter most to users.

15. In section 4.1.2, how to assess whether the AFDB-human models and the CATH-663 set are overlapped?

We apologise that this was not made clear in our original submission and have expanded the text in Methods section 'AFDB models used for fine-tuning' to give additional details on how overlap was determined.

16. In the last paragraph of section 4.1.2, it is unclear whether the 1195 AFDB-human models are a subset of the 7052 retained models. And how the 1195 AFDB-human models were selected?

We believe this has been addressed in point 15 above.

17. In the last paragraph of section 4.2, "In a final stage, the predicted domain assignments are postprocessed in a cleaning step which coalesces any domain with fewer than 30 residues or any segment fewer than 10 residues, with the domain preceding it." How to tell the domain of fewer than 30 residues and a segment of fewer than 10 residues?

The initial network output takes the form of an array consisting of domain indices, with one index assigned to each residue. This array can be tabulated, thereby determining the number of residues allocated to each domain. Any domains which contain fewer than a cut-off (30 residues), are coalesced into the previous domain.

For segments, we convert the array of domain indices into an array where domain indices are replaced with the length of the segment. For example:

The predicted domain index array:

[0,0,7,7,7,7,7,4,5,5,5,5,5,3,3]

Becomes:

[2,2,5,5,5,5,1,6,6,6,6,6,2,2]

This new array facilitates the look-up of segments which are under a certain threshold. We note that the minimum domain size and minimum segment length are user-defined parameters which can be adjusted for different use cases. In the future, with additional training data, we aim to improve on the architecture of Merizo to avoid the need for a cleaning step to be performed. We hope that this explanation has addressed the reviewer's query.

18. In the last paragraph of section 4.2, "When two domains that are distal to one another are assigned to the same domain index by the network (which, for example, can occur when a target contains a large number of domains), we separate these domains if the minimum distance between them is greater than 10Å."

If two domains are assigned the same domain index, indicating that all residues within these domains share the same domain index, it is unclear if they originate from distant domains. How can you tell they are from two distant domains? Do you go back and check the full-chain protein structures? If so, how do you check it? Does this mean you have to check all residues for each domain index for all proteins to see if they come from two or even more distant domains?

By "distal domains", we refer to instances where two or more domains have been assigned by the network to the same domain index, yet are physically separated by a Euclidean distance of over 10 Å (CA to CA). That is to say, that given the structure in question, that the two domains are not discontinuous domains.

The choice of a 10 Å threshold derives from a typical contact distance of 7-8 Å, with an additional buffer included to accommodate for minor model deviations. It is important to note that this

particular stage of the cleaning step exclusively occurs during inference on the full chain, and is not applied to the 512-residue crops employed during training.

This information was missing from our method description in our submission and we since added this to Methods section 'Network architecture'. We thank the reviewer for bringing up this important point.

Minor comments:

1. I highly recommend that the authors make the necessary datasets of Merizo, which comprise the training set, test set, as well as the ground truth for both sets, publicly available.

As requested, the datasets of Merizo (training and test sets) have been added to the Github repository at <https://github.com/psipred/Merizo/tree/main/datasets>.

2. In the introduction, what is "affinity learning approach"? I couldn't find a reference or more description of this method in the manuscript.

Affinity learning is a 'learning to cluster' strategy and is broadly explained in Supplementary Methods section 'Learning to cluster residues via embedding affinity'. We have added relevant references to the text in the introduction to give readers a broader overview than that used in our method.

3. In section 2.1, the authors stated that they downloaded data from CATH 4.2 to train the model. However, to my knowledge, CATH 4.3 was available by at least June 2021. Therefore, I wonder why the authors did not use the updated version in the study?

We apologise for this oversight – in fact, CATH 4.3 was the version used for this study. References to CATH 4.2 in our manuscript are incorrect and have been corrected.

4. In section 2.2, it is stated that "Merizo is a CATH-specific domain segmentation method". However, it is not clear why the authors specifically selected the CATH database as the ground truth dataset.

We thank the reviewer for this introspective question. This is a subjective matter, but as the reviewer is no doubt aware, the use of CATH is widespread in work concerning protein domain annotations. This includes all methods benchmarked against in our study. The use of CATH ground truths maintains comparability to these other methods. We also note that the corresponding author of this work is a founding member of CATH, described in the paper below:

Orengo, C.A., Michie, A.D., Jones, S., Jones, D.T., Swindells, M.B. and Thornton, J.M., 1997. CATH—a hierarchic classification of protein domain structures. Structure, 5(8), pp.1093-1109.

5. I recommend that the authors consider including the dataset name for each plot to enhance clarity. It can be challenging to refer to the dataset in the figures without this information. For instance, in Figure 3b, it would be beneficial to specify that the results are based on the CATH-663 dataset, while in Figure 3c, it should be mentioned that the prediction is related to the AFDB-1195 dataset. Without explicit dataset labels, these two plots may be mistakenly interpreted as representing the results of a single dataset.

This is a good suggestion and we have made the changes suggested by the reviewer. Where a dataset changes within a figure, we have added labels to clarify the origin of the data presented.

6. The claim made in the manuscript that “For several methods including Merizo, the average runtime per target is less than a second...” is not entirely correct. Merizo does not consistently achieve an average runtime of less than one second across different hardware. It is essential to include a description of the hardware used in the study as well.

Following the reviewer’s comment, we have made changes to the text to emphasize the hardware used for runtime measurements.

7. In section 4.2, the authors obtained the Euclidean backbone frames for each residue using the same algorithm in AlphaFold2. Could you explain what each dimension of [N,3,3], [N,3] represents? Other than N.

We have added a sentence to the end of paragraph 1 of Methods section ‘Network architecture’ which describes the dimensionality of the backbone frames.

8. “...512 residues. was carried out for...” It seems that “Training” is missed before “was”.

We thank the reviewer for spotting this error. The missing word is ‘Fine-tuning’ and we have amended this in the text.

Figure 1. Comparison of method performance against SCOPe and ECOD ground truths. a) Box plots show the IoU and MCC (± 20) distributions obtained by each method when using domain assignments from ECOD. Methods have been divided into ML-based (Merizo, Eguchi-CNN and DeepDom), non-ML-based (UniDoc and SWORD) and baseline (Random) methods. Data comprises 512 multi-domain targets, which are a subset of CATH-663. Solid white bars and white crosses represent distribution medians and means. Outliers are defined as data points exceeding 1.5IQR. **b)** Examples of segmentation outputs by ML methods (Eguchi-CNN, DeepDom and Merizo) and ground truths from ECOD and CATH. In each example, domains are shown in different colours. The number of domains as well as the IoU and MCC (± 20) scores obtained by each ML method is shown for each example. Ground truths from ECOD and CATH have been highlighted in orange boxes.

Figure 2. Domain assignments by UniDoc before and after pLDDT filtering. **a)** Example models and UniDoc assignments for i) AF-Q5VTH9-F1-model_v4, ii) AF-Q53SF7-F1-model_v4 and iii) AF-O60732-F1-model_v4. Left column shows the UniDoc assignment for the full chain model. Centre column shows the UniDoc assignment on the same structure, after a residue pLDDT of 60 is applied (residues with pLDDT < 60 are removed). Right column shows the UniDoc assignment at a pLDDT threshold of 70. Domains are shown in distinct colours and demarcated using letters A-D. **b)** Examples of domain damage by applying a pLDDT threshold of 60. Some residues are omitted in b) for clarity.

Figure 3. Examples of domain assignments from Merizo, before and after fine-tuning. Panels a) and b) correspond to the same models as shown in Figure 3d. In each example, the segmentation result after (left) and before (right) fine-tuning on NDRs is shown. Asterisks (*) highlight areas where domain assignments have been made to NDR regions. Domains have been demarcated using contrasting colours.

Figure 4. Examples of SCOPe domain annotations and assignments by Merizo and UniDoc. a) Examples of SCOPe domain boundaries which are part of secondary structure elements. The domain boundary is highlighted by a green box in each example. The inset in panel ii), highlights an internal chain break in the structure following the boundary SSE. b) The domain assignment of calmodulin from SCOPe, Merizo and UniDoc. In all examples shown, domains are coloured in distinct colours and NDRs are shown in white.

Reviewer #1 (Remarks to the Author):

I appreciate the thorough revisions made by the authors; the results align with my expectations. The only unexpected finding is that UniDoc's median MCC significantly surpasses that of Merizo, as shown in Figure 1 of the "response to review." However, Merizo demonstrates more consistent performance with a narrower distribution. Given the solid work in the revision, I recommend accepting the paper.

Reviewer #2 (Remarks to the Author):

The manuscript has improved a lot. Yet, there are still a few comments to be addressed. (The provided index is for reference to the first round of review.)

#8 (Major comments)

As I mentioned in my initial comment, there was not only a legend error in Figure 3b of the original submission.

1) The legend has been corrected in the revised version.

2) In Figure 3b (the current and previous submissions), the IoU distribution (after fine-tuning) corresponds to the MCC plot (to CATH) in Figure 2c, but it should actually correspond to the IoU plot in Figure 2c and Figure 2a (Merizo). The MCC distribution (after fine-tuning) in Figure 3c has the same problem. This means that the titles of two plots are reversed.

3) Furthermore, just switching the titles in Figure 3b cannot completely solve the problem in 2). This is because the IoU plot in Figure 3c should align with the MCC plot in Figure 2c, but it appears that the outliers of the distributions are not consistent.

Unfortunately, the issues mentioned in 2) and 3) have not yet been corrected.

New minor comments:

1. In the "Response to the reviewers": the sentence "Figure 1. Comparison of method performance against SCOPe and ECOD ground truths" should be "Figure 1. Comparison of method performance against ECOD ground truth" as there is no comparison with SCOPe in this figure.

Reviewer #1 (Remarks to the Author):

1. I appreciate the thorough revisions made by the authors; the results align with my expectations. The only unexpected finding is that UniDoc's median MCC significantly surpasses that of Merizo, as shown in Figure 1 of the "response to review." However, Merizo demonstrates more consistent performance with a narrower distribution. Given the solid work in the revision, I recommend accepting the paper.

Once again, we would like to thank the reviewer for taking the time to review our amended manuscript.

Reviewer #2 (Remarks to the Author):

The manuscript has improved a lot. Yet, there are still a few comments to be addressed. (The provided index is for reference to the first round of review.)

Major comments:

As I mentioned in my initial comment, there was not only a legend error in Figure 3b of the original submission.

1) The legend has been corrected in the revised version.

2) In Figure 3b (the current and previous submissions), the IoU distribution (after fine-tuning) corresponds to the MCC plot (to CATH) in Figure 2c, but it should actually correspond to the IoU plot in Figure 2c and Figure 2a (Merizo). The MCC distribution (after fine-tuning) in Figure 3c has the same problem. This means that the titles of two plots are reversed.

After further examination, the reviewer the correct and it appears that the 'IoU' and 'MCC (± 20)' axis labels have been erroneously swapped. This error only concerns Figure 3b and we have amended this in our resubmission.

3) Furthermore, just switching the titles in Figure 3b cannot completely solve the problem in 2). This is because the IoU plot in Figure 3c should align with the MCC plot in Figure 2c, but it appears that the outliers of the distributions are not consistent.

Unfortunately, the issues mentioned in 2) and 3) have not yet been corrected.

As mentioned above, it appears that the error could be remedied by swapping the axis labels for the subpanels of Figure 3b. Regarding the reviewer's comment that the outliers are inconsistent between the plots, we note that the shape of the outliers between Figures 2c and 3b for the post-finetuning results are vertically reflected between the plots. This however does not impact the validity of the results and only the visualisation. However, to make the connection between the plots clearer, we have reflected the outliers in Figure 3b to be consistent with those in Figure 2a and 2c. Furthermore, we have included a figure below (Figure 2 in this document) which isolates the panels in question (Figures 2a, 2c and 3b), showing them with minimal editing for comparison.

We would like to apologise that this error persisted through our latest resubmission and thank the reviewer for identifying it in the amended manuscript. We hope that the reviewer will agree that the error has now been addressed.

New minor comments:

1. In the "Response to the reviewers": the sentence "Figure 1. Comparison of method performance against SCOPe and ECOD ground truths" should be "Figure 1. Comparison of method performance against ECOD ground truth" as there is no comparison with SCOPe in this figure.

We apologise for this error and have included a revised version of the figure below.

Finally, we would like to thank the reviewer for their suggestions which we agree have strengthened our study.

Figure 1. Comparison of method performance against ECOD ground truth. a) Box plots show the IoU and MCC (± 20) distributions obtained by each method when using domain assignments from ECOD. Methods have been divided into ML-based (Merizo, Eguchi-CNN and DeepDom), non-ML-based (UniDoc and SWORD) and baseline (Random) methods. Data comprises 512 multi-domain targets, which are a subset of CATH-663. Solid white bars and white crosses represent distribution medians and means. Outliers are defined as data points exceeding 1.5IQR. **b)** Examples of segmentation outputs by ML methods (Eguchi-CNN, DeepDom and Merizo) and ground truths from ECOD and CATH. In each example, domains are shown in different colours. The number of domains as well as the IoU and MCC (± 20) scores obtained by each ML method is shown for each example. Ground truths from ECOD and CATH have been highlighted in orange boxes.

Figure 2. Comparison of box plot distributions for Figures 2a, 2c and 3b of the main document. Rows represent subpanels of figures, and columns represent IoU and MCC (± 20) distributions.

Reviewer #2 (Remarks to the Author):

The authors have addressed my comments.